# EXPANDYNERF:
# EXPANDING THE VIEWPOINT OF DYNAMIC SCENES BEYOND CONSTRAINED CAMERA MOTIONS

## ABSTRACT

In the domain of dynamic Neural Radiance Fields (NeRF) for novel view synthesis, current state-of-the-art (SOTA) techniques struggle when the camera's pose deviates significantly from the primary viewpoint, resulting in unstable and unrealistic outcomes. This paper introduces Expanded Dynamic NeRF (ExpanDyNeRF), a monocular NeRF method that integrates a Gaussian splatting prior to tackle novel view synthesis with large-angle rotations. ExpanDyNeRF employs a pseudo ground truth technique to optimize density and color features, which enables the generation of realistic scene reconstructions from challenging viewpoints. Additionally, we present the Synthetic Dynamic Multiview (SynDM) dataset, the first GTA V-based dynamic multiview dataset designed specifically for evaluating robust dynamic reconstruction from significantly shifted views. We evaluate our method quantitatively and qualitatively on both the SynDM dataset and the widely recognized NVIDIA dataset, comparing it against other SOTA methods for dynamic scene reconstruction. Our evaluation results demonstrate that our method achieves superior performance[1].

## 1 INTRODUCTION

Novel view synthesis plays a critical role in applications such as mixed reality (Xu et al., 2023; Gu et al., 2024), medical supervision (Yu et al., 2023; Wysocki et al., 2024), autonomous driving (Tancik et al., 2022; Zhang et al., 2024), wildlife observation (Zhang et al., 2023b; Sinha et al., 2023). Recent advances in Neural Radiance Field (NeRF) and its dynamic variants have significantly improved the efficacy of 3D scene reconstruction and novel view synthesis, achieving high precision (Wang et al., 2022b; 2024; Xu et al., 2024), speed (Garbin et al., 2021; Gao et al., 2024; Lee et al., 2024), and versatile style editing (Gu et al., 2021; Chen et al., 2024). On the other hand, Gaussian splatting based methods (Wu et al., 2023; 2024) offer a promising alternative with their efficient and flexible framework for high-quality rendering. Both methods render sharp and clear content from the primary perspectives of monocular inputs; however, novel view renderings often appear blurry and filled with artifacts, especially when significantly deviating from the primary camera view. This observation, shown in Fig. 1, is understandable since both methods lack supervision from diverse views while training, a limitation inherent to monocular camera settings.

To address this challenge, we introduce Expanded Dynamic NeRF (ExpanDyNeRF), an innovative method for dynamic 3D scene reconstruction. ExpanDyNeRF not only handles the primary camera view but also expands 3D reconstructions from significantly deviated views. Our end-to-end pipeline, illustrated in Fig. 2, includes a novel-view pseudo ground truth strategy that supervises the model optimization from novel views by leveraging Gaussian prior (Tang et al., 2023), which precisely defines dynamic object contours and color features across frames.

Moreover, in order to efficiently evaluate our method, we need a multi-view dataset with dynamic main camera motion and corresponding rotated side views. The commonly used NVIDIA dataset (Yoon et al., 2020) has dynamic camera motion but lacks ground truth for rotated side views. Other multi-view datasets such as the DyNeRF dataset (Li et al., 2022) have numerous rotated side views,

---

[1]The code and our data will be publicly available upon acceptance.

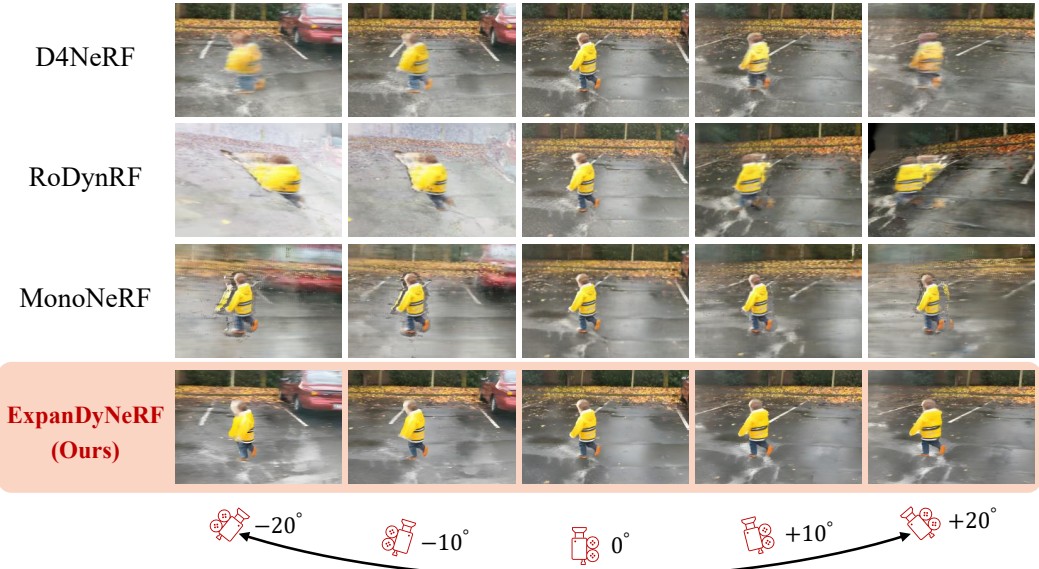

Figure 1: The figure highlights the shortcomings of leading dynamic NeRF models, such as D4NeRF Zhang et al. (2023a), RoDynRF Sabour et al. (2023), and MonoNeRF Tian et al. (2023), when rendering from new viewpoints distant from the original camera angle, exhibiting issues like blurring, distortion, and layering, with objects appearing flat, akin to cardboard cut-outs, at different camera angles. It showcases novel views rotated $10°$ and $20°$ to the left and right. Contrarily, the bottom row demonstrates the superior quality of novel view renderings achieved by our ExpanDyNeRF model, showcasing its capability to more accurately and reliably reconstruct dynamic scenes.

but no camera motion is introduced during recording. It is understandable that these limitations exist, considering that multi-view video recording with dynamic camera motion is difficult to collect in the real world due to the size and cost of the equipment. Therefore, to best demonstrate the efficacy of our method and provide reproducible data source for multi-angle rotation evaluation, we developed a Grand Theft Auto V (GTA V)-based Synthetic Dynamic Multi-view (SynDM) dataset (Rockstar Games, 2013). A unique dynamic camera dome system is designed for data collection. It guarantees camera motion for the main camera while providing corresponding side views for evaluation purposes. In this work, our evaluation primarily focuses on our SynDM dataset while only providing a qualitative comparison on the NVIDIA dataset. Our key contributions are summarized as:

- Identifying the deficiencies and inadequacies in the rendering results of SOTA monocular dynamic NeRF at largely deviated perspectives.
- Proposing ExpanDyNeRF, an innovative method capable of producing plausible novel view synthesis at largely deviated perspective.
- Creating the first GTA V based Synthetic Dynamic Multi-view (SynDM) dataset for evaluating rotated view synthesis. SynDM introduces both camera motion and the corresponding groundtruth for rotated side views.
- Conducting an in-depth assessment of SOTA dynamic NeRF models utilizing our dataset and NVIDIA dataset, this analysis highlights the specific hurdles of dynamic novel view rendering and showcases the superior performance of our method.

## 2 RELATED WORK

**NeRF-based Dynamic Novel View Synthesis.** NeRF algorithms have emerged as a powerful technique for high-quality 3D scene reconstruction from a sparse set of images. Original NeRF (Mildenhall et al., 2021) leverages a fully connected deep neural network to model the volumetric scene function. This function outputs the color and density for any given 3D point and viewing direction, enabling the synthesis of novel views through volume rendering techniques. NeRFs have

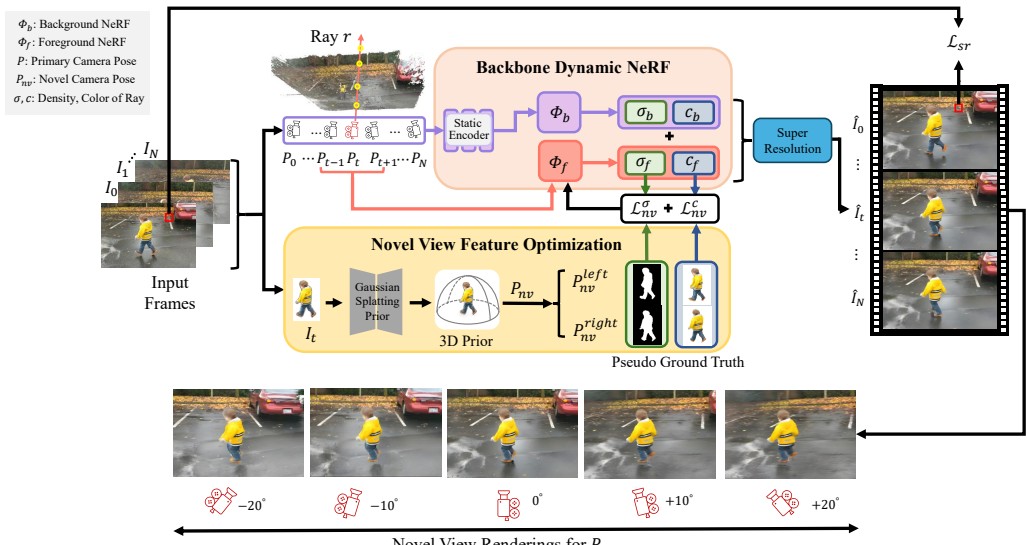

Figure 2: The ExpanDyNeRF architecture is structured into two main components: **(1) Backbone dynamic NeRF model** that processes rays to extract density ($\sigma$) and color ($c$) features from both background ($\Phi_b$) and foreground ($\Phi_f$) models, generating rendering predictions $\hat{I}_t$ from primary camera positions, supervised by the super-resolution loss $\mathcal{L}_{sr}$ **(2) Novel View Feature Optimization** uses the Gaussian Splatting based method to generate a 3D prior for each frame, facilitating the optimization of density and color features via pseudo ground truth for novel views. This includes updating $\sigma_f$ and $c_f$ using novel view loss metrics $\mathcal{L}_{nv}^{\sigma}$ and $\mathcal{L}_{nv}^{c}$, respectively, enhancing feature representation across different perspectives.

been particularly successful in static scenes, and recent advancements have extended their application to dynamic 3D reconstruction. Dynamic NeRF methods, such as HyperNeRF (Park et al., 2021), D4NeRF (Zhang et al., 2023a), and MonoNeRF (Fu et al., 2022) incorporate temporal components, allowing for the modeling of scenes with moving objects and varying illumination. These methods often employ additional strategies like temporal consistency loss and motion field modeling to handle the complexities of dynamic environments. Despite their success, dynamic NeRFs face challenges with computational expense and the need for densely sampled temporal data.

**Gaussian Splatting-based Dynamic Novel View Synthesis.** 3D Gaussian splatting (Kerbl et al., 2023), an alternative approach to 3D reconstruction, has gained popularity due to its computational efficiency and flexibility. This technique models the scene using a set of Gaussian blobs, each characterized by its mean, covariance, and color. These Gaussian blobs can be rendered efficiently with point-based rendering techniques, which makes this method well-suited for real-time applications. Recent advancements have extended Gaussian splatting to dynamic scenes using monocular inputs, enabling the capture of temporal changes from a single viewpoint. Notable methods like 4D Gaussian Splatting (4DGS) (Wu et al., 2023) and Deformable 3D Gaussian Splatting (Yang et al., 2023) focus on modeling deformable objects in dynamic scenes using monocular inputs. By incorporating deformation fields into the Gaussian representation, the method can capture complex non-rigid motions and surface deformations, providing a more detailed and accurate representation of the dynamic environment. The integration of monocular cues with deformable Gaussian splatting allows for robust and flexible reconstruction, bringing high resolution real-time rendering into the dynamic 3D reconstruction domain.

Both NeRF and Gaussian splatting-based approaches offer robust frameworks for dynamic 3D reconstruction, each with its own unique strengths and limitations. NeRF excels in high-fidelity reconstruction but is computationally intense, while Gaussian splatting provides a more efficient alternative with real-time, high-resolution capabilities. However, both methods encounter challenges in monocular settings, especially when rendering novel views from significantly deviated angles. In our method, we couple the strengths of both approaches with our own to address the current limitations of monocular 3D scene reconstruction.

## 3    METHOD: EXPANDYNERF MODEL & SYNDM DATASET

We present ExpanDyNeRF, a model capable of rendering dynamic 3D scenes from significantly deviated perspectives given monocular input. Additionally, we address the research gap related to the lack of ground truth for novel views of dynamic scenes captured by a monocular camera. The proposed ExpanDyNeRF model is elaborated in two parts: In Section 3.1, we introduce the dynamic NeRF structure that serves as the backbone of our ExpanDyNeRF model; In Section 3.2, we outline how we use the generated 3D Gaussian prior as pseudo-novel view ground truth and explain how this process supervises the optimization of the backbone model. Finally, in Section 3.3, we present the simulation setup and multi-view data acquisition method for our SynDM dataset.

### 3.1    EXPANDYNERF MODEL ARCHITECTURE

Following the structure of (Zhang et al., 2023a), we leverage two intertwined neural networks as our backbone NeRF model: $\Phi_b$ for the static background and $\Phi_f$ for the dynamic foreground. Details can be found in Algorithm 1 and Fig. 2

---

**Algorithm 1** ExpanDyNeRF Training Process

---

**Require:**

  Total training epoch $\mathbf{E}$. $N$ input frames $\mathbf{I}$. Ground truth camera poses $\mathbf{P}$. Epoch threshold $T$.
  Image-to-3D prior $F_{3D}$. Sampled rays $\mathbf{R}$. Sampled patch $Q$. Pretrained VGG-19 $F_{vgg}$.

**Ensure:**

  $\Phi_b, \Phi_f, \omega_b, \omega_f, \Theta$: Trained parameters.

1: $\Phi_b, \Phi_f, \omega_b, \omega_f, \Theta \leftarrow$ Initialize networks and weights.
2: **for** epoch $= 1 \rightarrow \mathbf{E}$ **do**
3:   **for** $I_t \in \mathbf{I}, t \in [1, N]$ **do**
4:     **for** $r \in \mathbf{R}$ **do**                           ▷ Sample rays from primary view at $t$
5:       $X, d \leftarrow \mathcal{F}(P_t)$                        ▷ Sample points along each ray
6:       $(\sigma_b, c_b), (\sigma_f, c_f) \leftarrow \Phi_b(X, d, \theta \sim P_\Theta(\theta)), \Phi_f(X, d, t)$       ▷ Query both modules
7:       $\hat{C} \leftarrow$ Integrate $(X, d, \omega_b, \omega_f)$       ▷ Integrate colors and densities of $X$ on each ray
8:     **end for**
9:     $\mathcal{L}_{cont} \leftarrow \sum \|\sigma_f(t+1) - \sigma_f(t)\|^2$                    ▷ Calculate continuity loss
10:    $\mathcal{L}_{rec} \leftarrow \sum \|\hat{C}(r) - C(r)\|_2^2$                       ▷ Calculate reconstruction loss
11:    $\mathcal{L}_{sr} \leftarrow \|\hat{Q} - Q\|_1 + \sum_l \lambda_l \|F_{vgg}^l(\hat{Q}) - F_{vgg}^l(Q)\|_1$     ▷ Calculate super-resolution loss
12:    **if** epoch $> T$ **then**
13:      $\mathbf{R}_{nv} \leftarrow$ Sample rays from $P_{nv}$                  ▷ Sample rays from novel view at $t$
14:      $\hat{C}(r_{nv}), \hat{\sigma}(r_{nv}) \leftarrow \Phi_f($ Sample points on $r_{nv} \in \mathbf{R}_{nv}, t)$       ▷ Features of $r_{nv}$ at $t$
15:      $C(r_{nv}), \sigma(r_{nv}) \leftarrow F_{3D}(I_t, r_{nv})$              ▷ Generate pseudo ground truth
16:      $\mathcal{L}_{nv} = \mathcal{L}_{nv}^c + \mathcal{L}_{nv}^\sigma \leftarrow \|\hat{C}(r_{nv}) - C(r_{nv})\|_2^2 + \|\hat{\sigma}(r_{nv}) - \sigma(r_{nv})\|_2^2$
17:      $\mathcal{L} \leftarrow \mathcal{L}_{cont} + \mathcal{L}_{rec} + \mathcal{L}_{sr} + \mathcal{L}_{nv}$
18:    **end if**
19:   **end for**
20:   Update $(\Phi_b, \Phi_f, \omega_b, \omega_f, \Theta)$ to minimize $\mathcal{L}$                 ▷ Update network parameters
21: **end for**
22: **return** $\Phi_b, \Phi_f$

---

**Preliminaries:** We utilize $N$ video frames $I_t, t \in [1, N]$, to reconstruct the scene's point cloud and estimate the primary camera poses $P$. For each pose $P_t \in P$, we calculate ray trajectories that allow us to sample points $\mathbf{x} = (x, y, z)$ along ray $r$ at time $t$ in direction $\mathbf{d}$. This process, represented by the function $\mathcal{F}(P_t) \rightarrow (\mathbf{x}, \mathbf{d})$, links the camera orientation to the sampled spatial points.

**Static Background Representation:** The static background module, $\Phi_b$, takes all $N$ frames as input and utilizes a distribution-based encoding to model the static elements in a scene. The distribution encoder enhances the alignment between camera projections and the static background distribution, simplifying the rendering process by focusing on minimal variations in the static features. This

module, defined by $\Phi_b(\mathcal{F}(P), \theta) \to (c_b, \sigma_b)$, predicts color $c_b$ and density $\sigma_b$ of spatial points from all poses in $P$, using a predefined distribution ($\theta \sim P_\Theta(\theta)$).

**Dynamic Foreground Representation:** The dynamic foreground component, $\Phi_f$, integrates spatial coordinates, viewing directions, and temporal inputs within a three-frame sliding window to capture and predict dynamic scene changes ($\Phi_f(\mathcal{F}(P_t), t) \to (c_f, \sigma_f)$). It achieves temporal consistency by encoding time $t$ within its inputs, ensuring the model is aware of temporal variations. Spatial-temporal consistency is achieved through optical flow-based scene flow estimations, which forecast the future states of dynamic elements. Additionally, continuity constraints are applied to maintain smooth transitions in attributes across frames, expressed as: $\mathcal{L}_{cont} = \sum \|\sigma_f(t+1) - \sigma_f(t)\|^2$.

**Transmittance Weight:** To seamlessly integrate static and dynamic representations at the ray level, the model uses learnable transmittance weights, $\omega_b$ for the background and $\omega_f$ for the foreground. These weights refine the rendering of transmittance values, allowing precise control over the blending of static and dynamic components during the rendering process. The rendering output is balanced according to the equation: $\hat{C}(r) = \omega_b \cdot \hat{C}_b(r) + \omega_f \cdot \hat{C}_f(r)$, where $\hat{C}_b$ and $\hat{C}_f$ represent the integrated color and density contributions from the background and foreground, respectively, over ray $r$.

**Primary View Reconstruction Loss:** The system employs a reconstruction loss to optimize $\Phi_b$ and $\Phi_f$ by minimizing the discrepancies between the features $\hat{C}(r)$ from rendered images and $C(r)$ from the ground truth images, defined as $\mathcal{L}_{rec} = \sum_{i=1}^{N} \sum_{r \in R} \|\hat{C}(r) - C(r)\|_2^2$. This loss ensures the renderings from the primary views closely match the ground truth frames, setting up a baseline for the following novel view optimization.

**Super-Resolution Loss:** Inspired by the state-of-the-art super-resolution techniques (Wang et al., 2021; 2022a), we introduce a super-resolution loss ($\mathcal{L}_{sr}$) to enhance image quality. The rendered patches from ExpanDyNeRF are processed by the pre-trained super-resolution model that upscales low-resolution inputs while preserving fine textures. The super-resolution loss is computed by sampling patches from the prediction and reference high-resolution images. The formula for $\mathcal{L}_{sr}$ is:

$$\mathcal{L}_{sr} = \sum_{k=1}^{K} \left\| \hat{Q}_k - Q_k \right\|_1 + \sum_{k=1}^{K} \sum_l \lambda_l \left\| F_{vgg}^l(\hat{Q}_k) - F_{vgg}^l(Q_k) \right\|_1$$

Here, $\hat{Q}_k$ and $Q_k$ represent the super-resolution prediction and reference patches, respectively, $F_{vgg}^l$ is a set of layers in a pretrained VGG-19 feature extractor, and $\lambda_l$ is the reciprocal of the number of neurons in layer $l$, combining reconstruction and perceptual losses.

## 3.2 PSEUDO GROUND TRUTH OPTIMIZATION STRATEGY

Through empirical experiments, we observed that foreground objects appear much blurrier than the background when the viewpoint rotates. This is due to affine effects, where objects closer to the viewpoint undergo more significant changes compared to those further away. Therefore, our method will prioritize foreground optimization. This method utilizes the Gaussian prior (Tang et al., 2023) to generate high-quality 3D priors, enabling the creation of a pseudo ground truth to supervise the model's optimization from new perspectives.

**Pseudo Ground Truth Generation for Novel Views**:

For each input frame $I_t$, we first construct a 3D Gaussian prior for the foreground object within the corresponding coordinate system. In this coordinate system, we establish a dome system centered on the foreground object, with a radius $R_d$ as shown in Fig. 3. The radius $R_d$ represents the distance from the primary viewpoint to the object. The position of $P_t$ on the dome is denoted as ($elevation = e, azimuth = 0, radius = R_d$), where $e$ corresponds to the elevation angle of the primary recording view (the rotation angle along the x-axis of $P_t$). We then sample novel viewpoints on the dome, maintaining the radius $R_d$ while varying the azimuth and elevation within a given range. The forward vector of the camera poses $P_{nv}^{(d)}$ at these novel viewpoints points towards the center of the dome. From these novel camera poses, we render pseudo ground truths that capture the density and color representations of the foreground object for each frame.

To supervise the novel view optimization with the pseudo ground truth, we first need to map the newly sampled novel view camera poses from the Gaussian prior coordinate system to our NeRF

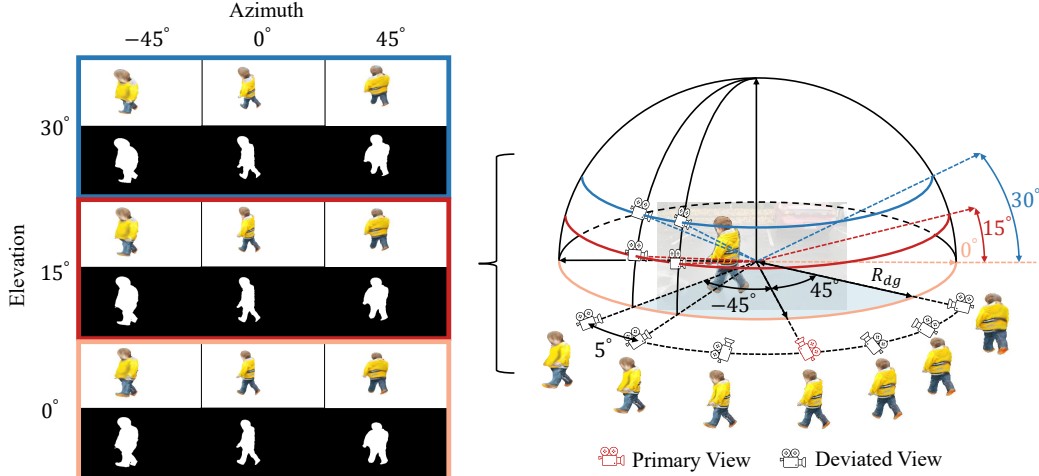

Figure 3: Generating pseudo ground truth for new viewpoints involves constructing a 3D model of the dynamic foreground object using the Gaussian Splatting technique and surrounding it with a dome centered on the foreground. The primary recording view is highlighted in red camera. Deviated novel viewpoints, shown in black, are created by rotating around the central point within a $R_{dg}$ radius, spanning from $-45°$ to $45°$ in $5°$ increments, at elevations of $0°$, $15°$, and $30°$. Examples of renderings and shape masks at azimuths $-45°$, $0°$, and $45°$ are provided for clarity.

coordinate system. Then, we compute the rendering predictions at those novel camera poses and calculate the loss between the predictions and the pseudo ground truth.

**Mapping Novel Views to the NeRF Coordinate System**: We first define the primary camera pose $P_t$ in the foreground NeRF coordinate system as $P_t^{(n)}$ and the corresponding one in Gaussian prior coordinate system as $P_t^{(d)}$. The transformation matrix $T$ that maps camera poses from the Gaussian prior to the foreground NeRF coordinate system can be calculated as $T = P_t^{(n)} \cdot (P_t^{(d)})^{-1}$. For the novel view camera poses $P_{nv}^{(d)}$ sampled in the Gaussian prior coordinate system, we use the transformation matrix $T$ to transfer all new camera positions to the foreground NeRF coordinate system, expressed as: $P_{nv} = \{P \cdot T, \forall P \in P_{nv}^{(d)}\}$.

**Novel View Loss**: During each model training iteration, two symmetrical novel views are randomly selected for each frame from $P_{nv}$. For these novel views, a set of rays $\mathbf{R}_{nv}$ is sampled from the camera pose. Color and density predictions in the foreground NeRF are derived from $\Phi_f(\mathcal{F}(P_{nv}), t)$, producing $(c_f, \sigma_f)$. Here, $\mathcal{F}(P_{nv}) \to (\mathbf{x}, \mathbf{d})$ samples points along a ray $r_{nv} \in \mathbf{R}_{nv}$. We then integrate the color and density values along $r_{nv}$ to obtain pixel-wise predictions $\hat{C}_f(r_{nv})$ and $\hat{\sigma}_f(r_{nv})$. The corresponding novel view loss is calculated as follows:

$$\mathcal{L}_{nv} = \mathcal{L}_{nv}^c + \mathcal{L}_{nv}^\sigma = \sum_{r \in R_{nv}} \left( \|\hat{C}(r_{nv}) - C(r_{nv})\|_2^2 + \|\hat{\sigma}(r_{nv}) - \sigma(r_{nv})\|_2^2 \right),$$

where $C(r_{nv})$ and $\sigma(r_{nv})$ represent the pseudo ground truth values for color and density on ray $r_{nv}$, respectively. This novel view loss is added to the total loss after specific epochs to manage the exploding gradient issue effectively. The final loss function is expressed as:

$$\mathcal{L} = \mathcal{L}_{cont} + \mathcal{L}_{rec} + \mathcal{L}_{sr} + \mathcal{L}_{nv}$$

### 3.3 SYNTHETIC DYNAMIC MULTIVIEW (SYNDM) DATASET

To demonstrate our method's efficacy and provide the essential ground truth for evaluating expanded novel views, we introduce our SynDM dataset. We leverage the high visual quality, open-world video game GTA V as the source platform for our dataset. Given that GTA V is limited to a single viewport, acquiring multi-view dynamic scenes presents a significant challenge. We expanded the GTAV-TeFS (Luo et al., 2023) method, a pioneering approach for generating dual-camera vision from the otherwise limited GTA V platform, to simultaneously support both monocular primary camera

capture and multi-view stereo camera collection in GTA V's dynamic and detailed environment. Traditionally, to enable multi-camera collection with only one viewport available, we need to perform frame swapping to display each camera view in sequence and repeat. This would inevitably lead to at least one frame latency (16.7ms) per swap under a 60 Hz refresh rate setting. This method remains valid for recording stationary scenarios and no additional camera motion should be introduced during the recording. However, in our scenario, all cameras are in motion, and the more cameras we add, the more latency we accumulate. Therefore, we designed a custom plugin to semi-freeze the game's graphic state while keeping the rendering engine and physics engine active during the controlled camera swap moment. With meticulous sequential planning, we were able to reduce the latency from 16.7ms per swap to 0.2ms per swap, making the resulting dataset depth-ready for all potential tasks.

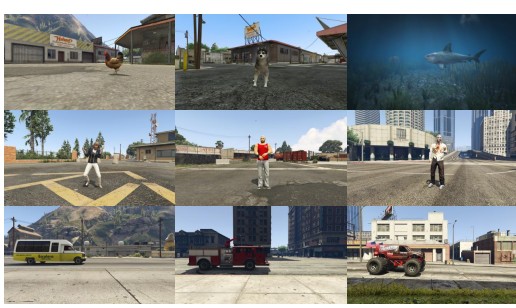

Figure 4: Gallery of images from our SynDM dataset, showcasing a variety of subjects. Animals are featured in the top row, humans in the middle row, and vehicles in the bottom row.

Our dataset supports synthetic object tracking via multi-view recordings, providing a comprehensive ground truth dataset for evaluating novel view synthesis within dynamic NeRF architectures. We collected nine distinct scenes under three main categories, each featuring dynamic entities such as humans, vehicles, and animals. Samples can be found in Fig. 4. For each scene, there are 19 cameras positioned horizontally around the reference point, following the similar design of Fig. 3, spaced at $5°$ intervals from $-45°$ to $45°$. An anchor camera is placed in the middle. On the vertical side, there are three more cameras elevated at the location of $-45°$, $0°$, and $45°$, respectively. In total, there are 22 cameras. The resolution for all images is 1920x1080, with a field of view of $90°$ horizontally and $59°$ vertically. The dataset is accompanied by metadata containing the camera position, camera rotation, character position, and reference point position.

## 4 EXPERIMENTAL RESULTS

In this section, we conduct a comprehensive comparison between our ExpanDyNeRF and four SOTA novel view synthesis methods: RoDynRF (Liu et al., 2023), MonoNeRF (Fu et al., 2022), 4DGS (Wu et al., 2023), and D4NeRF (Zhang et al., 2023a). All models are CC-By licensed. In Section 4.1, we describe the usage of our SynDM dataset and the NVIDIA dataset during the evaluation of our method. Section 4.2 shows the qualitative evaluation of ExpanDyNeRF on both the SynDM dataset and the NVIDIA dataset. In Section 4.3, we quantitatively demonstrate our model's capability in synthesizing deviated novel views superior to those of other SOTA methods. Finally, in Section 4.4, we perform an ablation study on different optimization strategies. Implementation details have been included in the supplementary material.

### 4.1 DATASETS

**SynDM Dataset**. In this experiment, we analyzed five different scenes from SynDM. They are Male, Female, Chicken, Dog, and Bus scenes. They showcase our method's performance in both rural and urban areas in the GTA V environment. We used the first 24 frames from each scene for training purpose. For evaluation, we generated 12 novel views for each frame of every scene from $-30°$ to $+30°$ in $5°$ intervals to compare with the ground truth.

**NVIDIA Dataset (Yoon et al., 2020)**. For qualitative evaluation with other SOTA models, we evaluated our method on the scenes captured using 12 stationary multi-view cameras. For quantitative evaluation, we selected 5 multi-view scenes with accurately estimated camera poses. Following the setup described in (Zhang et al., 2023a), we used 24 frames from the video footage for each scene in our experiments. The images were uniformly resized to a height of 272 for training purposes.

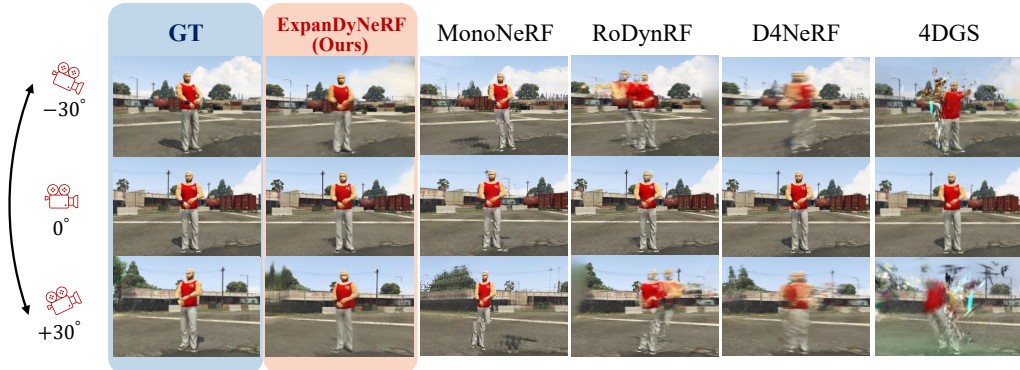

Figure 5: Comparison of dynamic NeRF models using novel view predictions on the SynDM dataset, showcasing ExpanDyNeRF and its counterparts. Each column represents a model's performance, with rows displaying views rotated by specific angles. The blue column presents the ground truth from SynDM at a corresponding angle. Red boxes highlight key performance differences among the models, with ExpanDyNeRF notably excelling in color and shape fidelity in novel view reconstructions.

## 4.2 QUALITATIVE EVALUATION

We present numerous visual comparisons on results from both SynDM and NVIDIA datasets, shown in Fig. 5 and Fig. S1, and more in supplementary (Fig. S6, Fig. S7, Fig. S8 and Fig. S9), respectively. It is evident that the novel view rendering ability of ExpanDyNeRF significantly surpasses other methods, particularly in maintaining the shape and color stability of dynamic parts. Although MonoNeRF shows clarity in rendering dynamic scenes from various angles, it struggles with conveying depth information, often resulting in a flat, cardboard-like appearance of objects. Furthermore, the background becomes significantly blurred in side views. Although 4DGS preserves background clarity, it struggles with inaccurate depth information in dynamic areas, causing objects to appear fractured upon rotation. This is particularly noticeable in the chicken scene shown in Fig. S6, where the entire chicken appears completely dispersed at different depths after a 30° camera rotation. Similarly, RoDynRF cannot accurately place the same object at consistent depths within the scene. In the tests using the NVIDIA dataset illustrated in Fig. S1, we observe that after rotation, the rendered figures only have their feet in the correct position, while the body sticks to the background pillars. Compared to our method, D4NeRF struggles to maintain the shape of dynamic parts due to the lack of supervision from side views. Overall, ExpanDyNeRF excels in maintaining shape and color stability in dynamic scenes, outperforming other methods that struggle with depth inaccuracies and fractured objects upon rotation.

Table 1: Quantitative comparison results on SynDM dataset. The best result is in bold and the second-best results are marked in blue.

| Method | Male | | | Female | | | Chicken | | |
|---|---|---|---|---|---|---|---|---|---|
| | FID↓ | PSNR↑ | LPIPS↓ | FID↓ | PSNR↑ | LPIPS↓ | FID↓ | PSNR↑ | LPIPS↓ |
| 4DGS (Wu et al., 2023) | 87.83 | 16.97 | 0.305 | 262.8 | 11.99 | 0.461 | 315.3 | 18.55 | 0.272 |
| RoDynRF (Liu et al., 2023) | 167.3 | 19.66 | 0.318 | 292.3 | 17.53 | 0.391 | 262.0 | 21.00 | 0.302 |
| MonoNeRF (Fu et al., 2022) | 178.5 | 17.46 | 0.441 | 312.2 | 15.35 | 0.572 | 287.3 | 15.11 | 0.545 |
| D4NeRF (Zhang et al., 2023a) | 144.4 | **22.80** | 0.388 | 168.0 | **19.81** | 0.448 | 290.6 | 22.69 | 0.378 |
| **ExpanDyNeRF (Ours)** | **66.52** | 22.16 | **0.144** | **77.66** | 19.05 | **0.173** | **155.8** | **23.66** | **0.142** |

| Method | Dog | | | Bus | | | Average | | |
|---|---|---|---|---|---|---|---|---|---|
| | FID↓ | PSNR↑ | LPIPS↓ | FID↓ | PSNR↑ | LPIPS↓ | FID↓ | PSNR↑ | LPIPS↓ |
| 4DGS (Wu et al., 2023) | 258.6 | 20.27 | 0.182 | 69.81 | 12.87 | 0.573 | 198.9 | 16.13 | 0.359 |
| RoDynRF (Liu et al., 2023) | 206.5 | 20.73 | 0.328 | 99.04 | 17.11 | 0.523 | 205.4 | 19.21 | 0.372 |
| MonoNeRF (Fu et al., 2022) | 260.5 | 20.07 | 0.370 | 77.39 | 16.24 | 0.601 | 223.2 | 16.85 | 0.506 |
| D4NeRF (Zhang et al., 2023a) | 148.3 | 23.05 | 0.394 | 111.4 | **18.43** | 0.555 | 172.5 | 21.36 | 0.433 |
| **ExpanDyNeRF (Ours)** | **67.35** | **23.74** | **0.151** | **43.78** | 17.64 | **0.294** | **82.22** | 21.25 | **0.219** |

| Ground Truth | $L_{nv}^{\sigma} + L_{nv}^{c} + L_{sr}$ | $L_{nv}^{\sigma} + L_{nv}^{c}$ | $L_{nv}^{\sigma}$ only | $L_{nv}^{c}$ only | Baseline |

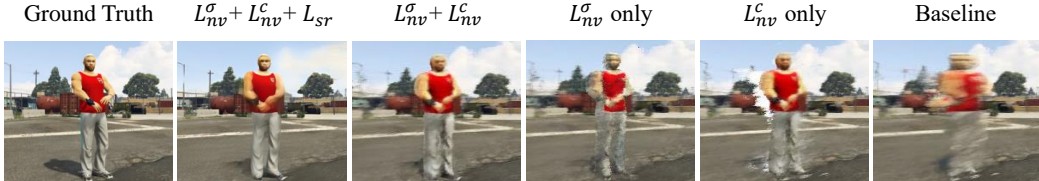

Figure 6: The images above, taken from a view rotated by -30 degrees, demonstrate the impact of different loss functions on the rendering quality in our novel view synthesis. The best performance is achieved when all losses are applied simultaneously, demonstrating the importance of each loss in improving rendering quality and matching the ground truth.

Table 2: Quantitative Evaluation of optimization strategies on the SynDM Dataset. 'Baseline' is without any optimization, '$L_{nv}^{\sigma}$' only uses density optimization, '$L_{nv}^{c}$' only applies color optimization, and '$L_{nv}^{\sigma} + L_{nv}^{c}$' trained with both. '$L_{nv}^{\sigma} + L_{nv}^{c} + L_{sr}$' is trained with all of module including super resolution.

| Method | Male | | | Female | | | Chicken | | |
|---|---|---|---|---|---|---|---|---|---|
| | FID↓ | PSNR↑ | LPIPS↓ | FID↓ | PSNR↑ | LPIPS↓ | FID↓ | PSNR↑ | LPIPS↓ |
| Baseline | 144.4 | **22.80** | 0.388 | 168.0 | 19.81 | 0.448 | 290.6 | 22.69 | 0.378 |
| $L_{nv}^{\sigma}$ | 128.2 | 22.43 | 0.393 | 147.1 | **19.90** | 0.452 | 237.8 | 23.57 | 0.374 |
| $L_{nv}^{c}$ | 118.0 | 21.96 | 0.392 | 162.8 | 19.88 | 0.442 | 315.3 | 23.35 | 0.378 |
| $L_{nv}^{\sigma} + L_{nv}^{c}$ | 113.4 | 21.96 | 0.322 | 138.4 | 19.66 | 0.428 | 207.9 | **23.89** | 0.339 |
| $L_{nv}^{\sigma} + L_{nv}^{c} + L_{sr}$ | **66.52** | 22.16 | **0.144** | **77.66** | 19.05 | **0.173** | **155.8** | 23.66 | **0.142** |

| Method | Dog | | | Bus | | | Average | | |
|---|---|---|---|---|---|---|---|---|---|
| | FID↓ | PSNR↑ | LPIPS↓ | FID↓ | PSNR↑ | LPIPS↓ | FID↓ | PSNR↑ | LPIPS↓ |
| Baseline | 148.3 | 23.05 | 0.394 | 111.4 | 18.43 | 0.555 | 172.5 | 21.36 | 0.433 |
| $L_{nv}^{\sigma}$ | 115.4 | 23.54 | 0.376 | 60.13 | **18.67** | 0.551 | 137.7 | **21.62** | 0.429 |
| $L_{nv}^{c}$ | 124.1 | 23.41 | 0.387 | 62.72 | 18.18 | 0.513 | 156.6 | 21.36 | 0.422 |
| $L_{nv}^{\sigma} + L_{nv}^{c}$ | 106.8 | **23.95** | 0.341 | 59.33 | 18.28 | 0.502 | 125.2 | 21.54 | 0.386 |
| $L_{nv}^{\sigma} + L_{nv}^{c} + L_{sr}$ | **67.35** | 23.74 | **0.151** | **43.78** | 17.64 | **0.294** | **82.22** | 21.25 | **0.219** |

## 4.3 QUANTITATIVE EVALUATION

We evaluate ExpanDyNeRF on five scenes from the SynDM dataset, focusing on PSNR, LPIPS, and FID metrics, which reflect the reconstruction quality, perceptual similarity, and distribution similarity, respectively–as seen in Table 1. In scenes such as "Male," our method achieves the second-best PSNR performance (which measures pixel-level reconstruction quality); however, it is important to note that D4NeRF also achieves relatively high PSNR scores despite producing heavily blurred images (Fig. 5), suggesting that PSNR may not be sensitive to certain distortions. Therefore, PSNR is not the most reliable metric for evaluating predictions with large deviation. In contrast, FID and LPIPS provide a more accurate assessment of image fidelity and perceptual similarity, where our model demonstrates superior performance compared to other methods. For instance, we achieve the lowest LPIPS scores (0.219, which is 40% lower than the second-best score of 0.359), indicating that our renderings are more perceptually similar to the ground truth. Moreover, the FID scores further highlight our model's effectiveness, delivering the best results (82.22 on average, more than twice as good as the second-best score of 172.5), showing that the distribution of ExpanDyNeRF's renderings aligns more closely with the ground truth compared to other methods.

## 4.4 ABLATION STUDY

To further validate the effectiveness of our proposed method, we conducted an ablation study focusing on varying optimization strategies. As shown in Fig. S5, the baseline result rendered using methods that lack optimization displays poorly defined shapes and noticeable blurring. Although results optimized with $L_{nv}^{c}$ show reduced blurring, the absence of shape optimization from $L_{nv}^{\sigma}$ during training leads to white artifacts around the objects. Models that utilize only $L_{nv}^{\sigma}$ maintain good shape integrity, but their colors appear to be faded or distorted due to insufficient color optimization from $L_{nv}^{c}$. Clear and stable rendering results are achieved when both $L_{nv}^{\sigma}$ and $L_{nv}^{c}$ are applied, although they lack the fine texture seen in the ground truth. Using color $L_{nv}^{c}$, density $L_{nv}^{\sigma}$ and super-resolution

$L_{sr}$ loss simultaneously yields the best results, closely matching the ground truth. This demonstrates that utilizing $L_{nv}$ can effectively eliminate artifacts, such as ghosting and blurring, that occur during novel view rotations.

The quantitative comparisons shown in Table 2 highlight the effectiveness of our method across various metrics. While PSNR scores show no conclusive trend due to their limitations for evaluating deviated angles as previously discussed, FID and LPIPS scores consistently improve with the application of novel view and super-resolution loss. The best results are achieved with the full combination of $L_{nv}^{\sigma} + L_{nv}^{c} + L_{sr}$. This demonstrates the effectiveness of our approach in enhancing the perceptual quality and reducing the distance between the distributions of generated and real images.

## 5 DISCUSSION AND CONCLUSION

**Limitations.** Although our experimental results demonstrate superior performance compared to other models, several limitations remain when viewed in a broader context. These include sub-optimal performance at wider viewing angles (greater than 45 degrees of deviation) and unsatisfactory visual results when generating previously unseen background information. Addressing these challenges will be a key focus of future research, with the goal of improving performance in extreme angles and enhancing the realism of unseen scene generation.

**Conclusion.** ExpanDyNeRF advances dynamic NeRF by significantly improving novel view synthesis, particularly at wider viewing angles, by extending the range of stable visualization. Our SynDM dataset, based on GTA V for dynamic multiview scenarios, provides a strong foundation for evaluating dynamic scene reconstructions from varied angles. Our evaluations demonstrate ExpanDyNeRF's superior ability to render dynamic scenes.

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

# A    Supplementary Materials

**Implementation Details.** In our comparative experiments, we employed the settings provided in the papers for each method. Our proposed model was trained on 2 V100 graphics cards, with a total of 600,000 iterations for each scene. The weight coefficients for the three losses we introduced, $L_{nv}^{c}$, $L_{nv}^{\sigma}$ and $L_{sr}$, are 1.0, 0.1 and 0.5 respectively. All other parameter settings were kept consistent with baseline (Zhang et al., 2023a). Additionally, the ray sampling strategy we used involved sampling within a bounding box with a padding of 2.

Table S1: Quantitative comparison results on NVIDIA dynamic scenes. The best result is in bold, and the second-best results are marked in blue.

| Method | Truck PSNR↑ | Truck LPIPS↓ | Umberalla PSNR↑ | Umberalla LPIPS↓ | Playground PSNR↑ | Playground LPIPS↓ | Balloon1 PSNR↑ | Balloon1 LPIPS↓ | Balloon2 PSNR↑ | Balloon2 LPIPS↓ | Average PSNR↑ | Average LPIPS↓ |
|---|---|---|---|---|---|---|---|---|---|---|---|---|
| DynNeRF (Gao et al., 2021) | 25.78 | 0.134 | 23.15 | 0.146 | 23.65 | 0.093 | 21.47 | 0.125 | 25.97 | 0.059 | 24.00 | 0.111 |
| NeRF (Mildenhall et al., 2021) | 27.93 | 0.098 | 21.23 | 0.234 | 20.75 | 0.157 | 21.41 | 0.141 | 23.30 | 0.071 | 22.92 | 0.140 |
| 4DGS (Wu et al., 2023) | 26.39 | 0.239 | 23.17 | 0.280 | 19.43 | 0.268 | 23.31 | 0.235 | 25.75 | 0.238 | 23.61 | 0.252 |
| RoDynRF (Liu et al., 2023) | 29.13 | 0.063 | **24.26** | **0.089** | **24.96** | **0.048** | 22.37 | 0.103 | 26.19 | 0.054 | 25.38 | 0.071 |
| Mononerf (Fu et al., 2022) | 27.56 | 0.115 | 23.62 | 0.180 | 22.61 | 0.130 | 21.89 | 0.129 | 27.36 | 0.052 | 24.61 | 0.121 |
| D4NeRF (Zhang et al., 2023a) | **31.75** | **0.041** | 24.20 | 0.104 | 23.94 | 0.073 | **23.87** | **0.067** | **27.60** | **0.041** | **26.27** | **0.065** |
| ExpanDyNeRF (Ours) | 30.60 | 0.034 | 23.71 | 0.152 | 22.06 | 0.129 | 23.79 | 0.076 | 27.41 | 0.050 | 25.50 | 0.088 |

**Quantitative Evaluation on NVIDIA Dataset.** In Table S1, we conducted a quantitative analysis of our method on the NVIDIA dataset and compared it with several current state-of-the-art methods. The results are satisfactory. Although we did not achieve the highest results in the tests, this observation stems from the unique properties of the NVIDIA dataset. The NVIDIA dataset was captured using a system of 12 stationary cameras. Therefore, the camera poses used in the training and test sets are the same, even though the video frames tested are from different times. These 12 cameras are almost all positioned directly in front of the scene, leading to superior test results for these methods on the NVIDIA dataset. However, because we further optimized for side views of the scene, it resulted in a decline in test results for the primary views (camera poses in the NVIDIA dataset). **Qualitative Evaluation on SynDM and NVIDIA Dataset.** The novel view predictions showcase

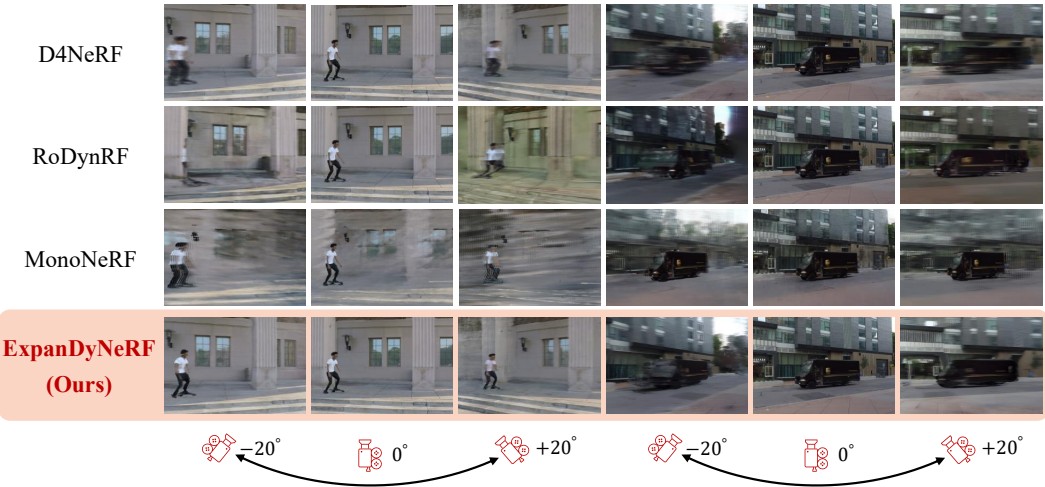

Figure S1: Demonstration of the rendering capabilities of leading dynamic NeRF models in new camera perspectives, using the NVIDIA dataset with skating and truck sequences for illustration. It displays the original video capture viewpoint and views rotated $20°$ to the left and right around the center of the foreground for each sequence. The outcomes indicate that our ExpanDyNeRF model exhibits minimal distortion in color and shape.

the performances of the state-of-the-art dynamic NeRF models alongside our ExpanDyNeRF in the following figures, all trained on the SynDM dataset and NVIDIA dataset. It is evident that ExpanDyNeRF significantly surpasses others in terms of novel view rendering stability, particularly in maintaining the shape and color stability of dynamic parts.

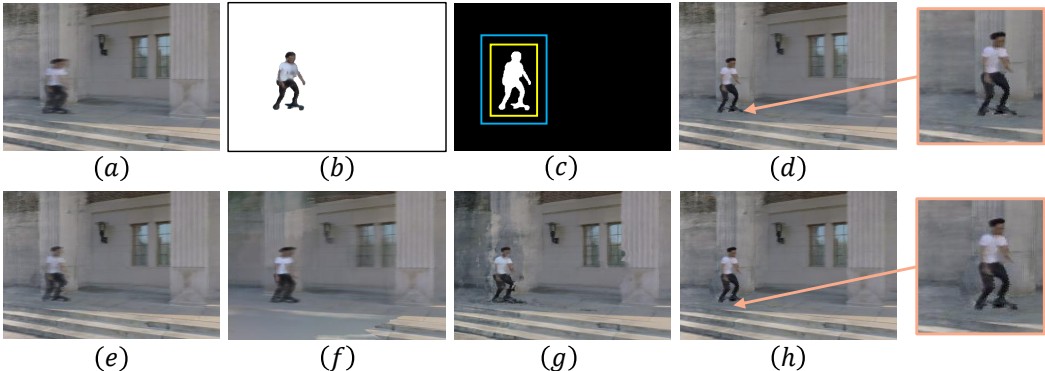

Figure S2: This figure presents an ablation study on our ray sampling strategy. Panel $(a)$ displays the base model's output without optimization for color and density. Panel $(b)$ depicts the pseudo-ground-truth of novel views from the created 3D mesh. Panel $(c)$ illustrates the density mask derived from pseudo ground truth, where the yellow and blue boxes represent the bounding box with 2-pixel padding, and 10-pixel padding, respectively. Panel $(e)$ shows predictions from global ray sampling on the mask, while panel $(f)$ shows predictions from ray sampling within the foreground object area only. Panel $(g)$ demonstrates the GaussianBlur strategy's prediction. Panels $(d)$ and $(h)$ showcase predictions with 2-pixel and 10-pixel padding, respectively. The comparison between panels $(d)$ and $(h)$ reveals that employing 2-pixel padding leads to enhanced quality in reconstructing novel view details with minimum background distortion.

**Ray Sampling Strategies** We compared various ray sampling strategies for novel view density and color optimization in Equation 3.2. Examples are shown in Fig. S2. Global sampling over the whole frame yields results in Panel $(e)$ similar to the base output in Panel $(a)$, due to the small proportion of dynamic segments in the frame, causing generalized and ineffective updates. Alternate strategies sample within the foreground object's area shown white in Panel $(c)$, which may overlook updates outside this zone. Panel $(f)$ demonstrates that sampling from various viewpoints for dynamic density updates can unintentionally extend beyond the intended mask, causing non-dynamic areas to obscure the background. Panel $(g)$ shows the third strategy where the GaussianBlur (Gonzalez, 2009) expands the foreground boundary, creating a zero gray-scale edge. Sampling within this blurred mask improves results, yet areas adjacent to the person still see undue dynamic density updates beyond the motion mask. Our final strategies focused on ray sampling within the padded area of the mask's bounding box (bounding boxes in Panel $(c)$), which outperforms the other strategies. Experimentation showed that while larger padding, like 10 pixels in Panel $(h)$, achieves comparable foreground optimization to smaller padding, such as 2 pixels in Panel $(d)$, it adversely affects background clarity.

Table S2: This table compares key attributes across various popular datasets for dynamic 3D reconstruction, highlighting the proposed SynDM dataset that outperforms other datasets by uniquely enabling quantitative evaluation of dynamic scene reconstruction from multiple deviated angles at any frame, a capability absent in the other datasets. The columns represent: (1) Multi-view: whether the dataset provides images from multiple viewpoints instead of monocular images per frame; (2) Deviated View GT: whether the dataset includes ground truth data from multiple deviated viewing angles (small and large deviations) per frame; (3) Unconstrained Scene: whether the dataset covers diverse recording locations (near and far) or object categories (human, animal, vehicle, etc.); (4) Cams Motion: whether the dataset was collected using a moving camera; and (5) Background: whether the dataset includes the full scene with both foreground and background.

| Dataset | Multi-view | Deviated View GT | Unconstrained Scene | Cams Motion | Background |
|---|---|---|---|---|---|
| DAVIS (Pont-Tuset et al., 2017) | ✗ | ✗ | ✓ | ✓ | ✓ |
| iPhone (Gao et al., 2022) | ✗ | ✗ | ✓ | ✓ | ✓ |
| NeRFDS (Yan et al., 2023) | ✗ | ✗ | ✓ | ✓ | ✓ |
| NVIDIA (Yoon et al., 2020) | ✗ | ✗ | ✓ | ✓ | ✓ |
| HyperNeRF (Park et al., 2021) | ✗ | ✗ | ✗ | ✓ | ✓ |
| DyNeRF (Li et al., 2022) | ✓ | ✓ | ✗ | ✗ | ✓ |
| ActorsHQ (Işık et al., 2023) | ✓ | ✓ | ✗ | ✗ | ✗ |
| Multi-face (Wuu et al., 2022) | ✓ | ✓ | ✗ | ✗ | ✗ |
| **SynDM(Ours)** | ✓ | ✓ | ✓ | ✓ | ✓ |

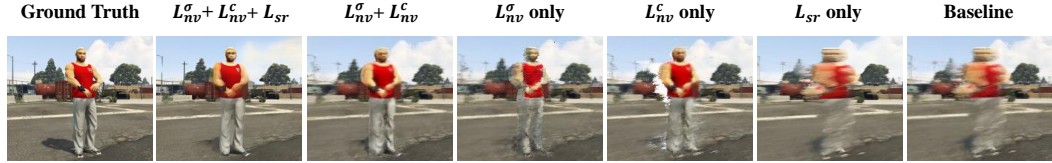

| Ground Truth | $L_{nv}^{\sigma} + L_{nv}^{c} + L_{sr}$ | $L_{nv}^{\sigma} + L_{nv}^{c}$ | $L_{nv}^{\sigma}$ only | $L_{nv}^{c}$ only | $L_{sr}$ only | Baseline |

Figure S3: The images above, rendered from a view rotated by -30 degrees, illustrate the impact of different loss functions on the quality of novel view synthesis. The best performance is achieved when all loss functions ($L_{nov} + L_{hw} + L_{sr}$) are applied simultaneously, highlighting the complementary role each loss plays in enhancing rendering quality and achieving a closer match to the ground truth. Notably, using the super-resolution loss ($L_{sr}$) alone does not improve rendering quality, as the significant gains observed in Table 2 are primarily due to the novel view loss ($L_{nov}$) effectively addressing the blurriness issue. This improvement provides the necessary foundation for the super-resolution loss to contribute meaningfully to the final rendering quality.

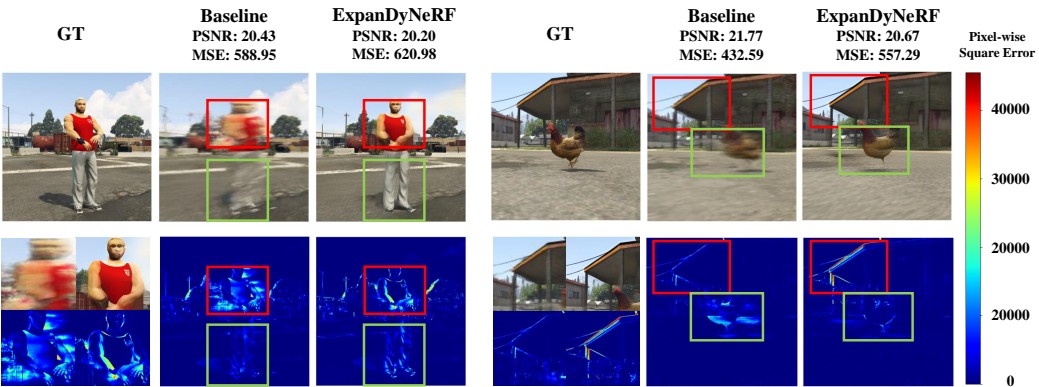

Figure S4: Comparison of visual and quantitative results between the baseline and ExpanDyNeRF models, evaluated using PSNR and MSE metrics. The "GT" column represents the ground truth images. The red and green boxes highlight critical regions of interest for analysis. The red box demonstrates areas with sharp and detailed reconstruction by ExpanDyNeRF, whereas the baseline exhibits significant blur. Despite ExpanDyNeRF producing clearer outputs, PSNR scores are lower due to localized high-density errors (highlighted in the pixel-wise error heatmap). In contrast, the baseline's blurry results yield smoother transitions, leading to lower MSE despite poorer visual quality. The green box further illustrates how blurry regions (e.g., human pants or chicken body) blend into the background, minimizing MSE contributions. This demonstrates the limitation of PSNR in capturing perceptual quality, particularly when evaluating sharpness and clarity.

**Necessity and Advantage of Proposed SynDM Dataset** Table S2 underscores the critical limitations of existing datasets for evaluating dynamic scene reconstruction under large deviations. None of the datasets simultaneously provide the essential features required for fair frame-wise quantitative evaluation, such as multi-view data, deviated view ground truth (GT), dynamic full-scene representation (including foreground and background), and camera motion. For instance, datasets like DAVIS, iPhone, and NVIDIA lack multi-view data, while NeRFDS, HyperNeRF, and DyNeRF offer multi-view information but fail to include deviated GT, which is crucial for robust quantitative metrics such as PSNR and LPIPS. Furthermore, many datasets, such as NeRFDS and DyNeRF, exclude background information, limiting their applicability for full-scene dynamic rendering. This absence of deviated GT across all existing real-world datasets severely restricts the ability of current methods, including ExpanDyNeRF, to perform frame-wise quantitative evaluations under large deviations. In contrast, our proposed SynDM dataset uniquely addresses these gaps by providing multi-view data, deviated GT, dynamic full-scene representation, and camera motion, enabling comprehensive evaluations that were previously unattainable. This highlights the indispensable role of SynDM in advancing dynamic scene reconstruction research and further underscores the need for a real-world dataset with comparable features, such as one we plan to develop using a large-scale moving camera dome with precise pose capture capabilities.

**Instant-NGP**  **3DGS**

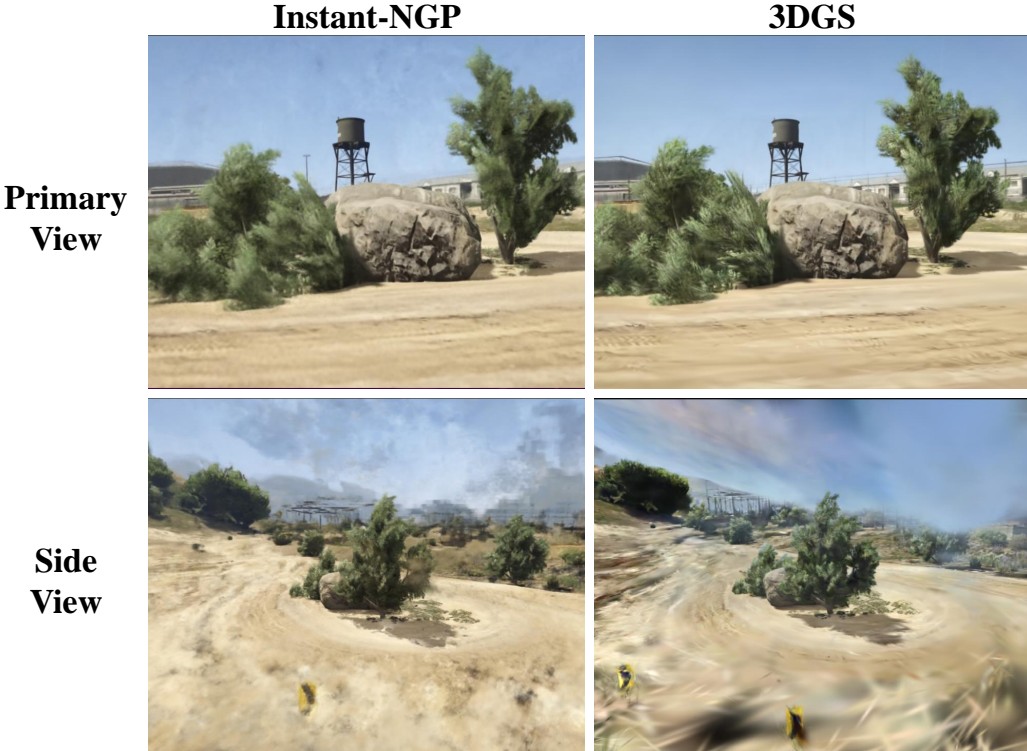

Figure S5: Comparison of scene reconstructions between Instant-NGP Müller et al. (2022) and 3DGS Kerbl et al. (2023) from both primary and side views. The primary view (top row) for both methods appears visually consistent, with the sky maintaining its expected uniformity, giving the impression of accurate reconstruction. However, in the side view, 3DGS (bottom right) introduces significant artifacts, with the sky being incorrectly reconstructed as a nearby structure, obstructing the distant background, including mountains and bushes. In contrast, Instant-NGP (bottom left) retains the expected characteristics of the sky as distant and uniform, achieving higher fidelity and richer scene details. These differences highlight the limitations of GS models in handling objects without clear geometric boundaries or at infinite distances, as opposed to NeRF-based methods.

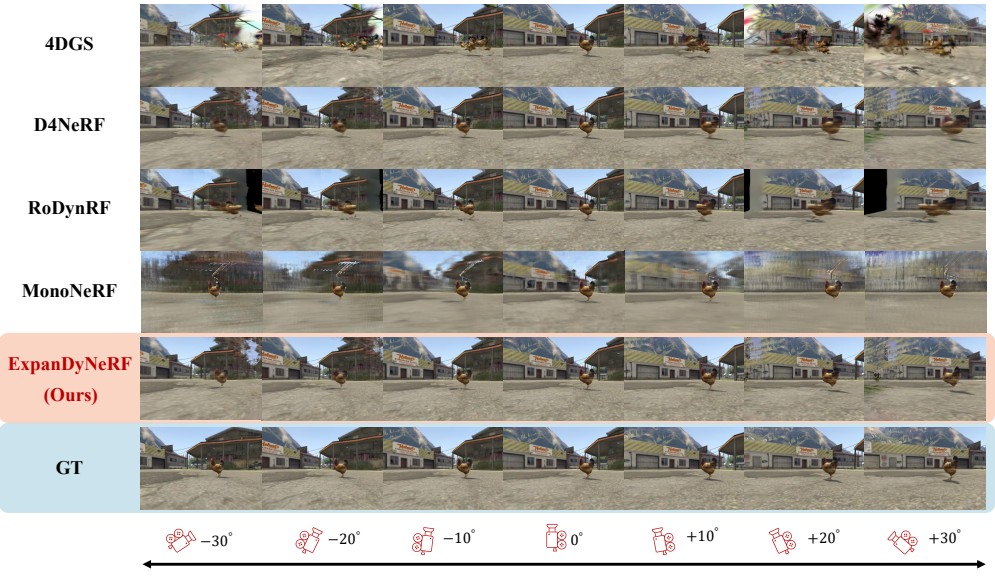

Figure S6: This figure presents an ablation study on the novel view synthesis performance of leading dynamic NeRF models and our ExpanDyNeRF training on the animal data from our SynDM dataset.

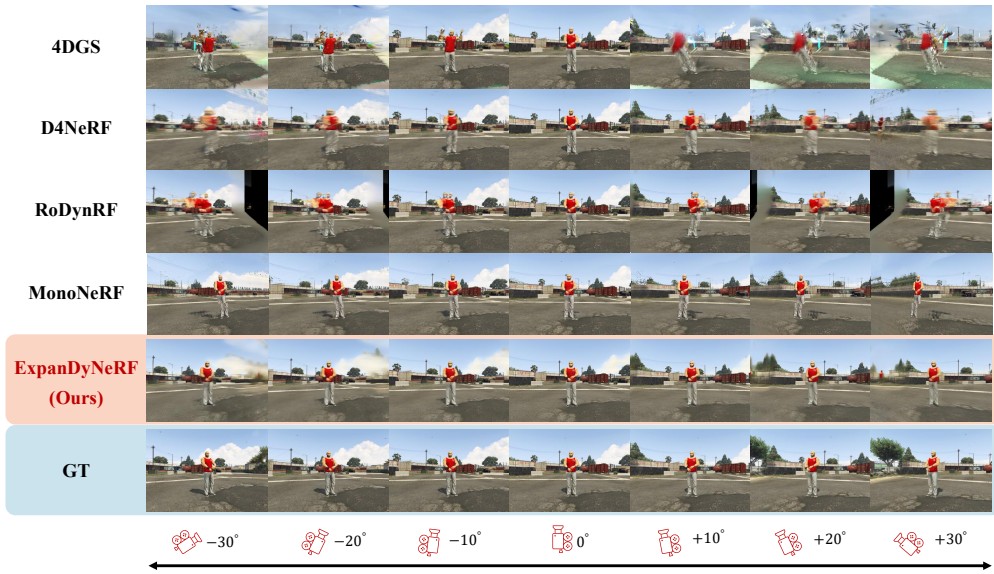

Figure S7: This figure presents an ablation study on the novel view synthesis performance of leading dynamic NeRF models and our ExpanDyNeRF training on the human data from our SynDM dataset.

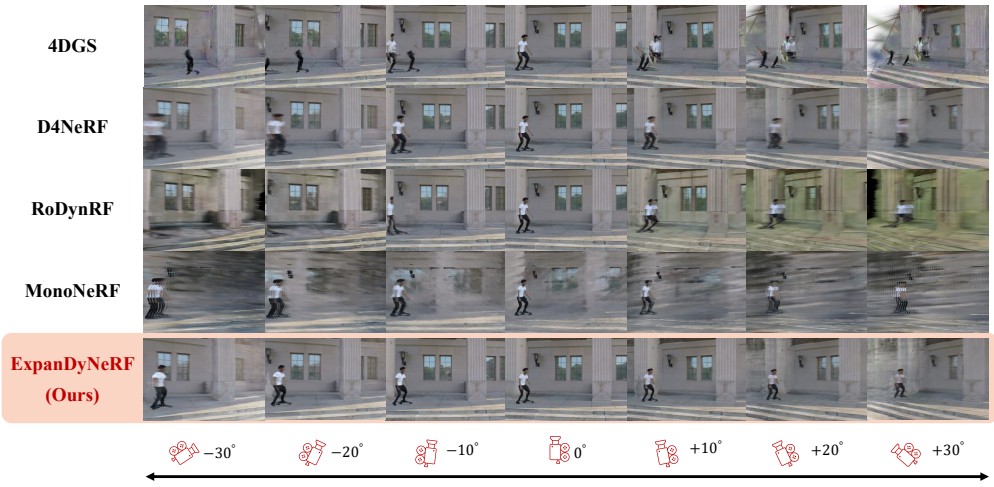

Figure S8: This figure presents an ablation study on the novel view synthesis performance of leading dynamic NeRF models and our ExpanDyNeRF training on the truck data from the NVIDIA dataset.

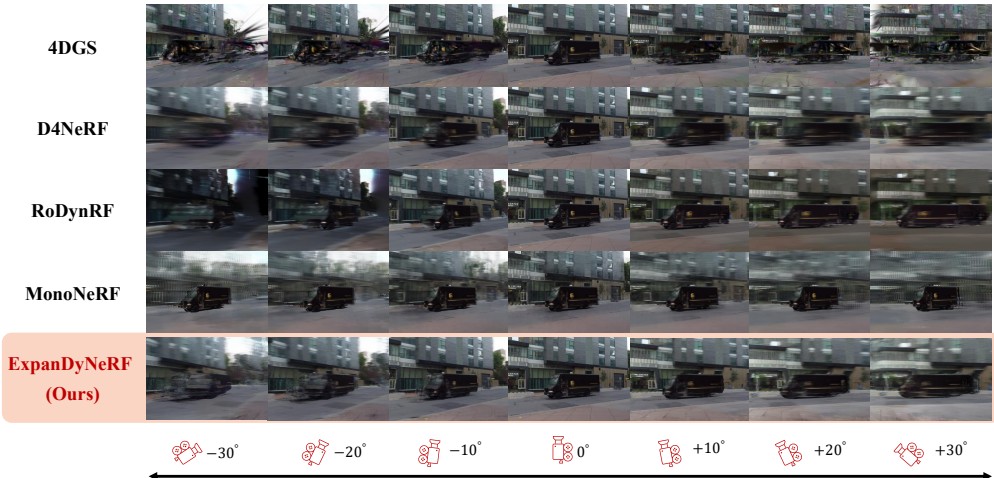

Figure S9: This figure presents an ablation study on the novel view synthesis performance of leading dynamic NeRF models and our ExpanDyNeRF training on the skating data from the NVIDIA dataset.

