# OpenReview forum: "ExpanDyNeRF: Expanding the Viewpoint of Dynamic Scenes beyond Constrained Camera Motions"
_ICLR.cc/2025/Conference — ICLR 2025 Conference Withdrawn Submission_

### Official Review · Reviewer_Vz1G · 2024-10-23

**Soundness:** 2
**Presentation:** 2
**Contribution:** 1
**Rating:** 3
**Confidence:** 4

**Summary:**

The authors proposed a new way to improve the performance of novel view synthesis (NVS) in dynamic scenes. They leverage the existing work $D^{4}NeRF$[1] for their scene optimization part. Besides, they utilize the DreamGaussian[2] to generate the 3D priors and then use the primary camera poses are supervision to fit the transformation between cameras and generated objects. In this way, novel views with novel cameras of the object are generated. However, I do think this paper is more likely to be a technical work instead of a research paper, as it lacks contributions in the method part. The good thing is they proposed a new dataset that can help the research in this field. Also, I think this paper should include more experiments with proper baselines and more public datasets to prove their effectiveness.

[1] Zhang, Boyu, Wenbo Xu, Zheng Zhu, and Guan Huang. "Detachable novel views synthesis of dynamic scenes using distribution-driven neural radiance fields." arXiv preprint arXiv:2301.00411 (2023).

[2] Tang, Jiaxiang, Jiawei Ren, Hang Zhou, Ziwei Liu, and Gang Zeng. "Dreamgaussian: Generative gaussian splatting for efficient 3d content creation." arXiv preprint arXiv:2309.16653 (2023).

**Strengths:**

1. The authors proposed a new dataset for the NVS task.
2. The authors conducted enough ablation studies to prove the contribution of each component of their model.
3. The author combine the GS system and NeRF system together.

**Weaknesses:**

1. The presentation of Algorithm 1 is really bad. From my perspective, the authors should explain the content in the Algorithm table instead of only 'Details can be found in ...', if they want to put this big table in the main paper.
2. I think the content in Sec 3.1 should be a high-level explanation of the existing work the authors followed.
3. I think the 'primary view' should be some prior camera knowledge, which means that the authors indeed feed prior cameras into their model and generate more novel views serving as more supervision. However, the baselines to which the authors compare their method were not given camera priors like RoDynRF,...
4. This paper is lacking of experiments to prove their effectiveness. I would like to see more comparisons on more public datasets (NeRF-DS[1], DAVIS[2], Hypernerf[3], iPhone[4], ...) with more recent works (spacetimeGS[5],... )

[1] Yan, Zhiwen, Chen Li, and Gim Hee Lee. "Nerf-ds: Neural radiance fields for dynamic specular objects." In Proceedings of the IEEE/CVF Conference on Computer Vision and Pattern Recognition, pp. 8285-8295. 2023.

[2] Pont-Tuset, Jordi, Federico Perazzi, Sergi Caelles, Pablo Arbeláez, Alex Sorkine-Hornung, and Luc Van Gool. "The 2017 davis challenge on video object segmentation." arXiv preprint arXiv:1704.00675 (2017).

[3] Park, Keunhong, Utkarsh Sinha, Peter Hedman, Jonathan T. Barron, Sofien Bouaziz, Dan B. Goldman, Ricardo Martin-Brualla, and Steven M. Seitz. "Hypernerf: A higher-dimensional representation for topologically varying neural radiance fields." arXiv preprint arXiv:2106.13228 (2021).

[4] Gao, Hang, Ruilong Li, Shubham Tulsiani, Bryan Russell, and Angjoo Kanazawa. "Monocular dynamic view synthesis: A reality check." Advances in Neural Information Processing Systems 35 (2022): 33768-33780.
[5] Li, Zhan, Zhang Chen, Zhong Li, and Yi Xu. "Spacetime gaussian feature splatting for real-time dynamic view synthesis." In Proceedings of the IEEE/CVF Conference on Computer Vision and Pattern Recognition, pp. 8508-8520. 2024.

**Questions:**

1. How do you decompose the static background and the dynamic foreground? Do you have G.T. motion masks?
2. Which type of distribution-based encoding network did you use? If from existing works, please cite them, if not, please claim clearer that you propose it.
3. Why do the authors leverage the Gaussian priors, but then map them to the NeRF system? Why not use the Gaussian system for all?
4. How long does the proposed method require per scene?

---

> ### Author Response · Authors · 2024-11-19
>
> Thank you for your valuable feedback and bringing up the questions. We will address your concerns and questions accordingly.
>
> **Weakness Concern 1: "The presentation of Algorithm 1 is really bad. From my perspective, the authors should explain the content in the Algorithm table instead of only 'Details can be found in ...', if they want to put this big table in the main paper."**
>
> --------------------
>
> Thank you for your feedback on the presentation of Algorithm 1. We understand your concerns and appreciate the opportunity to clarify. Including an algorithm table to present the high-level logic of a method is a common practice in ICLR papers. For example, similar approaches have been adopted in Section 3 of Salvatori et al. [1] and Section 3.2 of Laskin et al. [2], where phrases like "Details can be found in..." are used to direct readers to additional explanations.
>
> In our paper, the structure and logic of the method section are closely aligned with the content of Algorithm 1. Every formula and variable in the paper directly corresponds to elements in the table, ensuring there is no ambiguity or missing explanation. The algorithm table serves as a concise summary of the methodology, while the accompanying sections provide detailed elaborations on each step. We believe this format balances clarity and conciseness, which is particularly important given the space constraints in the main paper.
>
> We hope this addresses your concern, but we’re happy to refine the explanation further if you have specific suggestions. Thank you again for your thoughtful input!
>
> [1] Salvatori, Tommaso, et al. "A Stable, Fast, and Fully Automatic Learning Algorithm for Predictive Coding Networks." The Twelfth International Conference on Learning Representations.
>
> [2] Laskin, Michael, et al. "In-context reinforcement learning with algorithm distillation." arXiv preprint arXiv:2210.14215 (2022).
>
> **Weakness Concern 2: "I think the content in Sec 3.1 should be a high-level explanation of the existing work the authors followed."**
>
> ---------------------
>
> In Section 3.1, we focused on introducing the basic structure and methodology of D4NeRF, which serves as our baseline. While we didn’t explicitly frame it with phrasing like "D4 did … in … way," we aimed to provide a clear and concise explanation of the essential concepts and their relevance to our approach. We have attributed all referenced methods appropriately, such as in the opening of Section 3.1, where we state, "following the structure of [Zhang et al., 2023a]."
>
> The content in Section 3.1 is already high-level and specifically concentrated on explaining D4NeRF in a way that connects directly to our contributions. Making it even more high-level would risk oversimplifying the explanation and potentially losing clarity. However, we’re happy to refine the framing further to better align with your expectations if you have specific suggestions. Thank you again for your thoughtful review!
>
>
> **Weakness Concern 3: "I think the 'primary view' should be some prior camera knowledge, which means that the authors indeed feed prior cameras into their model and generate more novel views serving as more supervision. However, the baselines to which the authors compare their method were not given camera priors like RoDynRF,..."**
>
> ------------------------
>
> The primary view refers exclusively to the monocular video captured from a single camera at 0 degrees, as shown in Figure 3. All the compared methods are provided with the exact same input frames and the same camera poses (if the "camera prior" you’re referring to is the camera pose).
>
> Regarding RoDynRF, as you specifically mentioned, it has two versions: a pose-free version and another that uses ground-truth poses. To ensure the fairest comparison, we provide the same input frames and camera poses to RoDynRF, so it does not need to recalculate the poses itself. This approach is consistently applied to all the other models as well.
>
> Based on this setup, we believe our experimental comparisons are unbiased and ensure fairness across all evaluated methods.

---

> > ### Comment · Reviewer_Vz1G · 2024-11-23
> > **Response to Authors' Rebuttal**
> >
> > I appreciate the authors' response. However, most of my concerns are still valid as follows.
> > 1) In the Method section, Sec 3.1 mostly contains high-level descriptions of the existing works. Sec 3.3 includes the illustration of the proposed new dataset. So, I am still concerned that this work is not contributing enough on the method level because it is more likely to be some technical steps in Sec 3.2.
> > 2) Like other reviewers, I still think this paper does not have enough experiments to support its statements. Besides, I believe datasets like the iPhone, NeRF-DS, and Nvidia contain multi-view frames with large viewpoint changes at each timestamp. Please carefully refer to them. I understand the author claims that '**rendering at highly deviated angles**', but I believe it should not affect the renderings at the normal angles. Also, the authors do not make any clear statement about what is highly deviated angle. I keep my point that this paper lacks enough experiments.
> > 3) It lengthy running time poses the biggest limitation to this work.
> > 4) I do not agree with the authors' statement that '**GS-based methods struggle in this realm.**'.

---

> ### Author Response · Authors · 2024-11-19
>
> **Weakness Concern 4: "This paper is lacking of experiments to prove their effectiveness. I would like to see more comparisons on more public datasets (NeRF-DS[1], DAVIS[2], Hypernerf[3], iPhone[4], ...) with more recent works (spacetimeGS[5],... )"**
>
> ---------------------
>
> It is important to note that it is not a matter of us choosing whether or not to use these datasets, but rather that they simply do not meet the necessary conditions for a fair comparison with our method.
>
> To reiterate, our method is designed to address the challenge of full-scene rendering at highly deviated angles. This requires datasets with the following three critical characteristics:
>
> 1.	Monocular Input with Motion: The primary front-facing camera must capture motion, as this is essential for generating COLMAP results, which our method relies on as input.
>
> 2.	Multi-view with Side-view Ground Truth (GT): Each primary front-facing camera frame should be paired with significantly deviated side-view images, serving as ground truth for quantitative metrics such as PSNR and LPIPS.
>
> 3.	Dynamic Full-Scene Representation: Since our method renders the entire scene, including dynamic elements, the dataset must also contain background information.
>
> None of the existing available datasets meet all three criteria detailed above. Even the NVIDIA dataset we tested in the paper lacks the side view GT, which limits our ability to provide quantitative comparisons and restricts us to qualitative assessments.
>
> To fully outline this, we provide a comparison table below  (as well as Table S2 in the updated supplementary section). It is evident that none of the existing datasets—whether monocular or multi-view—simultaneously fulfill all three conditions. This limitation underscores the need for a new dataset, which is why we developed SynDM. In fact, SynDM and our work was specifically designed to address this gap and provide a strong foundation for future research in this domain.
>
> +-------------------+-----------------+---------------------------+---------------------------------+---------------------+--------------------+
>
> |----Dataset----|-Multi-view-|-Deviated-View GT-|-Unconstrained Scene-|-Cams Motion-|-Background-|
>
> +-------------------+-----------------+---------------------------+---------------------------------+---------------------+--------------------+
>
> |SynDM(Ours)|-------✔-------|-------------✔------------|----------------✔---------------|---------✔---------|---------✔---------|
>
> |-----DAVIS------|-------✘-------|-------------✘-----------|----------------✔---------------|---------✔---------|---------✔---------|
>
> |-----iPhone----|-------✘-------|-------------✘------------|----------------✔---------------|---------✔---------|---------✔---------|
>
> |----NeRFDS----|-------✘-------|-------------✘-----------|----------------✔----------------|---------✔---------|---------✔--------|
>
> |-----NVIDIA----|-------✘-------|-------------✘------------|----------------✔---------------|---------✔---------|---------✔--------|
>
> |--HyperNeRF-|-------✘-------|-------------✘------------|----------------✘---------------|---------✔---------|---------✔--------|
>
> |----DyNeRF----|-------✔-------|-------------✔------------|----------------✘---------------|---------✘---------|---------✔--------|
>
> |---ActorsHQ--|-------✔-------|-------------✔------------|----------------✘---------------|---------✘---------|---------✘--------|
>
> |---Multi face--|-------✔-------|-------------✔------------|----------------✘---------------|---------✘---------|---------✘--------|
>
> +-------------------+-----------------+---------------------------+---------------------------------+---------------------+--------------------+

---

> > ### Author Response · Authors · 2024-11-19
> >
> > **Q1: " How do you decompose the static background and the dynamic foreground? Do you have G.T. motion masks?"**
> >
> > ------------------------
> >
> > We utilize the standard Mask R-CNN [1] to generate motion masks for each frame. This approach is common practice in dynamic NeRF methods, such as D4NeRF and MonoNeRF, and has proven to be effective for separating dynamic foreground elements from the background.
> >
> > [1] He, Kaiming, et al. "Mask r-cnn." Proceedings of the IEEE international conference on computer vision. 2017.
> >
> > **Q2: "Which type of distribution-based encoding network did you use? If from existing works, please cite them, if not, please claim clearer that you propose it."**
> >
> > --------------------------
> >
> > We used the same distribution-based encoding network as in D4NeRF, which internally operates as a standard attention-based MLP. As stated in section 3.1, we utilized the foreground-background NeRF framework introduced in D4NeRF and provided the appropriate citation for it.
> >
> > **Q3: "Why do the authors leverage the Gaussian priors, but then map them to the NeRF system? Why not use the Gaussian system for all?"**
> >
> > We leverage the Gaussian prior in NeRF because of its fast and accurate foreground reconstruction which enhances foreground training for NeRF under deviated views. NeRF then provides us with superior reconstruction accuracy for outdoor scenes, especially for distant element like sky, which exist at infinite distances in our SynDM dataset. GS-based methods struggle in this realm, resulting in poor sky reconstruction in our preliminary exploration. GS methods are optimized for indoor scenes and objects without background and face challenges with infinite-distance elements. This limitation makes them unsuitable for our application, which focuses on full scene rendering in outdoor environments.
> >
> > This being said, we do recognize the potential advantages of an all-encompassing GS framework in reducing reconstruction time, and we are actively exploring solutions to leverage our SynDM dataset for further research in this direction.
> >
> >
> > **Q4: "How long does the proposed method require per scene?"**
> >
> > ---------------------
> >
> > ExpanDyNeRF requires approximately 72 hours to complete 600,000 iterations for a 24-frame input video for an entire training pipeline (including pseudo ground truth generation and target scene training), using 2 V100. In comparison, our baseline model, D4NeRF, takes 60 hours to train the same scene.

---

> ### Author Response · Authors · 2024-11-28
>
> Thank you for your valuable feedback and bringing up the questions. We will address your concerns and questions accordingly.
>
> **1. In the Method section, Sec 3.1 mostly contains high-level descriptions of the existing works. Sec 3.3 includes the illustration of the proposed new dataset. So, I am still concerned that this work is not contributing enough on the method level because it is more likely to be some technical steps in Sec 3.2.**
>
> We believe that evaluating the contribution solely based on the structure of the paper is an unreasonable approach. Moreover, challenge identification and dataset synthesis are also significant contributions of this work. Our study focuses on reconstructing dynamic scenes from viewpoints significantly different from the primary view, using only monocular input and without imposing constraints such as circular recording camera movements. We are the first to highlight this critical yet overlooked challenge in the field of dynamic scene reconstruction and to propose a robust solution. Unlike SOTA methods that primarily emphasize the accuracy of primary view reconstruction, our approach addresses this broader issue, as demonstrated by the results in Figures 1 and 5, which underscore our substantial contribution to this task. Notably, the dataset synthesis involves both theoretical and technical innovations, which are not achievable with the existing functionalities of GTA V alone.
>
>
> **2. I believe datasets like the iPhone, NeRF-DS, and Nvidia contain multi-view frames with large viewpoint changes at each timestamp. Please carefully refer to them. I understand the author claims that 'rendering at highly deviated angles', but I believe it should not affect the renderings at the normal angles. Also, the authors do not make any clear statement about what is highly deviated angle. I keep my point that this paper lacks enough experiments.**
>
> We have already tested our method on the NVIDIA Dynamic Dataset, with results presented in the supplementary material (Figures S1, S7, and S8), as referenced in Section 4.2. However, the NVIDIA dataset was captured using a handheld camera matrix where all cameras were located on a flat shelf and facing the same direction. Therefore, there are no significant viewing angle variations, only differences in position. This limitation prevents us from obtaining ground truth for side views.
>
> We investigated monocular video datasets such as iPhone and NeRF-DS before designing our own dataset. However, we found that these datasets lack multiview frames with significant viewpoint changes at each timestamp due to their inherently monocular nature. Initially, we considered using a camera pose from a frame with a temporal interval from the given frame as the novel view. However, this approach proved ineffective, as it fails to provide valid ground truth for novel views in dynamic environments. Objects in such scenes are constantly moving, so frames captured at later times—even from different viewpoints—do not accurately represent the same scene at the same moment as the original frame. Using frames from a larger temporal interval as ground truth for side views introduces temporal inconsistencies, ultimately compromising reconstruction accuracy.
>
> Regarding large, deviated angles, we define them in our paper as ranging from -45° to 45°, which exceeds the typical range found in datasets like DyNeRF, where angles are usually confined to -30° to 30°. This broader range demonstrates the robustness of our dataset and method. While we considered simulating a monocular video with camera motion using multiview datasets like DyNeRF by selecting frames from different cameras across consecutive time steps, this approach was debatable. The deviated views introduced during this imitation process would provide additional information during training, leading to unfair comparisons. Our analysis of cameras in DyNeRF, such as cam0, cam4, and cam5—relatively close to the front view—revealed that their deviation range is approximately [-10°, +10°]. However, our study shows that deviations exceeding 10° result in obvious blur and artifacts (Figure S7, S8), which means the simulated monocular training video from DyNeRF provide deviated view ground truth during training. To avoid potential criticisms for using side-view data to train a monocular-based method and to better showcase the advantages of our approach that can reconstruct the scene without the information from side views, we focused solely on the NVIDIA dataset, where all camera are aligned and have no deviation between cameras, and the SynDM dataset, where the camera pose deviation between adjacent frames is under 1°. Additionally, DyNeRF’s design does not enable precise control over deviation angles (e.g., 5°, 10°, 15°) for testing purposes. As a result, it is challenging to systematically analyze how the degree of deviation affects the rendering clarity.

---

> > ### Author Response · Authors · 2024-11-28
> >
> > **3. It lengthy running time poses the biggest limitation to this work.**
> >
> > We acknowledge that training speed is a common challenge in dynamic NeRF research. For instance, DynIBaR: Neural Dynamic Image-Based Rendering (CVPR 2023) [1] reported training times of approximately 48 hours, as stated in Section 4, Implementation Details: "Optimizing a full system on a 10-second video takes around two days using 8 Nvidia A100 GPUs." Similarly, MonoNeRF: Learning a Generalizable Dynamic Radiance Field from Monocular Videos (CVPR 2023) [2] required approximately 50 hours of training on a single Nvidia A100 GPU.
> >
> > While training efficiency is undoubtedly important for real-time applications, ExpanDyNeRF is NOT designed for real-time usage at this stage. Instead, the primary contribution of our work lies in proposing a novel approach for expanded-view video reconstruction. This has significant benefits for many offline applications, such as enhancing the field of view of monocular recordings in entertainment or educational settings.
> >
> > In short, the speed of our method is highly dependent on the hardware used for training. Based on GPU benchmark data from Lambda Labs (\url{https://lambdalabs.com/blog/nvidia-a100-vs-v100-benchmarks}), a single A100 GPU is 2.17 times faster than a V100, while 8 A100 GPUs are 15.81 times faster than a single V100. Considering that our system uses V100 GPUs, our training speed is well within a reasonable and competitive range for dynamic NeRF research.
> >
> > [1] Li, Zhengqi, et al. "Dynibar: Neural dynamic image-based rendering." Proceedings of the IEEE/CVF Conference on Computer Vision and Pattern Recognition. 2023.
> >
> > [2] Tian, Fengrui, Shaoyi Du, and Yueqi Duan. "Mononerf: Learning a generalizable dynamic radiance field from monocular videos." Proceedings of the IEEE/CVF International Conference on Computer Vision. 2023.
> >
> >
> > **4. I do not agree with the authors' statement that 'GS-based methods struggle in this realm.'.**
> >
> > To further prove our statement on GS model struggles on infinite distance object such as sky, we would like to provide additional clarification and evidence.
> >
> > In our preliminary experiments, we observed that 3DGS models struggle to reconstruct infinite distance objects accurately, particularly the sky. While the front-facing view appears relatively normal, side views reveal significant artifacts, such as the sky manifesting as a large, unnatural blue blob close to the scene geometry. This indicates that the model fails to preserve the spatial coherence and infinite distance properties of the sky during reconstruction.
> >
> > We include two example images (Figure S5 in the Appendix at the end of our paper) to substantiate our observations:
> >
> > 1.	Primary View (top row): For both GS and NeRF, the sky appears normal, creating an impression that the reconstruction is effective when viewed directly.
> >
> > 2.	Side View from 3DGS [3] (bottom right): The sky becomes highly distorted, losing its expected characteristics of being far away and uniform. Instead, it is incorrectly reconstructed as a nearby floating structure behind the tree and blocks the far away background scene (e.g. mountains and bushes), which detracts from the overall scene consistency.
> >
> > 3.	Side View from Instant ngp [4] (bottom left): Continuing with the NeRF perspective, the sky retains its proper distance, maintaining the expected characteristics of being far away and uniform. Additionally, the scene exhibits higher fidelity, with richer and more accurate details, enhancing overall visual coherence.
> >
> > These artifacts arise due to the inability of regular GS models to manage objects that lack distinct geometric boundaries or those at infinite distances, as the model fundamentally relies on discrete splatting in bounded spatial regions. To address this, we utilized NeRF as the backbone for scene reconstruction. Furthermore, we have already begun exploring approaches to integrate our proposed method into a full GS framework while addressing these limitations.
> >
> > [3] Kerbl, Bernhard, et al. "3D Gaussian Splatting for Real-Time Radiance Field Rendering." ACM Trans. Graph. 42.4 (2023): 139-1.
> >
> > [4] Müller, Thomas, et al. "Instant neural graphics primitives with a multiresolution hash encoding." ACM transactions on graphics (TOG) 41.4 (2022): 1-15.

---

### Official Review · Reviewer_rrER · 2024-10-28

**Soundness:** 3
**Presentation:** 3
**Contribution:** 2
**Rating:** 3
**Confidence:** 3

**Summary:**

The work proposes ExpanDyNeRF, a method for dynamic scene reconstruction and novel view synthesis, which handles significant camera rotational changes. Such robustness is achieved due to the integration of a Gaussian Splatting prior based on a generative image-to-3D model. In addition, a dynamic multi-view dataset based on the GTA-V video game SynDM is proposed to evaluate the rendering quality of dynamic subjects from significant viewing angle deviations from the primary camera and demonstrate the superior performance of ExpanDyNeRF in comparison to dynamic NVS baselines.

**Strengths:**

1. The paper is well-written and easy-to-follow.
2. The effort behind creating a multi-view GTA-V-based benchmark with a latency decrease from 16.7 ms to as low as 0.2 ms is impressive.
3. Ablation studies show the effectiveness of introduced novel view loss at largely deviated perspectives.

**Weaknesses:**

1. The proposed supervision requires training a Gaussian Splatting (GS) for every timestamp along with the dynamic NeRF model, which appears computationally suboptimal. With this, it would be important to emphasize the advantages of the proposed method in comparison to existing alternatives, such as DreamGaussian4D [1], with additional background reconstruction using a GS model.
2. Experimental results are rather limited (two datasets including a proposed benchmark) and do not demonstrate a clear advantage of the proposed method compared to other dynamic reconstruction methods. Moreover, none of the chosen baselines utilizes a 3D prior, which seems unfair as the current solution leverages one.
3. Operational range [-45$^\circ$, 45$^\circ$] is rather limiting, especially for 360$^\circ$ object-centric dynamic reconstruction for creation of avatars and 4D assets. Addressing the issue appears important for the practical usage of this work.

[1] Ren et al. "Dreamgaussian4d: Generative 4d gaussian splatting.", 2023.

**Questions:**

1. The concept of parallax described in Section 3.2 and motion perception are also addressed in [2] by leveraging semantic understanding. It would be interesting to include it as a baseline and test the method on the proposed significant rotational changes.
2. Can translational changes be addressed in the work, especially in the generation of novel deviated views? What about changes in the distance to the object (e.g. zoom-ins)?
3. Another potential benchmark could be ActorsHQ [3] or Multiface [4] for rendering dynamic humans and faces from various angles.


[2] Liu et al. "Gear-NeRF: Free-Viewpoint Rendering and Tracking with Motion-aware Spatio-Temporal Sampling.", 2024.
[3] Işık et al. "Humanrf: High-fidelity neural radiance fields for humans in motion.", 2023.
[4] Wuu et al. "Multiface: A dataset for neural face rendering.", 2022.

---

> ### Author Response · Authors · 2024-11-19
>
> Thank you for your valuable feedback and bring up the questions. We will address your concerns and questions accordingly.
>
> **Weakness Concern 1.1: "The proposed supervision requires training a Gaussian Splatting (GS) for every timestamp along with the dynamic NeRF model, which appears computationally suboptimal. "**
>
> -------------------------
>
> The computational cost of generating Gaussian Splatting (GS) per-frame is approximately 30 seconds. The entire process, for an input scene with 24 frames, takes around 15 minutes. This is negligible compared to the 60–72 hours required to train our dynamic NeRF model. We chose GS for its ability to achieve precise foreground object reconstruction efficiently. After evaluating SOTA methods like DreamGaussian [1], Stable-Zero123 [2], and One-2-3-45 [3], we selected DreamGaussian for its superior balance of accuracy and time efficiency.
>
> [1] Tang, Jiaxiang, et al. "Dreamgaussian: Generative gaussian splatting for efficient 3d content creation." arXiv preprint arXiv:2309.16653 (2023).
>
> [2] Liu, Ruoshi, et al. "Zero-1-to-3: Zero-shot one image to 3d object." Proceedings of the IEEE/CVF international conference on computer vision. 2023.
>
> [3] Liu, Minghua, et al. "One-2-3-45: Any single image to 3d mesh in 45 seconds without per-shape optimization." Advances in Neural Information Processing Systems 36 (2024).
>
>
> **Weakness Concern 1.2: "With this, it would be important to emphasize the advantages of the proposed method in comparison to existing alternatives, such as DreamGaussian4D [1], with additional background reconstruction using a GS model."**
>
> --------------------
>
> We utilize NeRF instead of a full Gaussian-Splatting framework because of its superior reconstruction accuracy for outdoor scenes, especially for distant elements like the sky, which exist at infinite distances in our SynDM dataset. GS often struggle in this regard, resulting in poor background reconstruction in our preliminary exploration. In fact, most GS methods are optimized for indoor scenes or objects without background and face challenges with infinite-distance elements. This limitation makes them unsuitable for our application, which focuses on full scene rendering in outdoor environments.
>
> This being said, we do recognize the potential advantages of an all-encompassing GS framework in reducing reconstruction time, and we are actively exploring solutions to leverage our SynDM dataset for further research in this direction.

---

> ### Author Response · Authors · 2024-11-19
>
> **Weakness Concern 2.1: "Experimental results are rather limited (two datasets including a proposed benchmark) "**
>
> ---------------------------
>
> We want to emphasize that none of the currently well-established datasets fulfill our requirements for evaluation. This is precisely what has motivated us to create the SynDM dataset and fill this gap in the field.
>
> To reiterate, our method is designed to address the challenge of full scene rendering at largely deviated angles. This requires datasets with the following three key characteristics:
>
> 1.	Monocular input with motion: The primary front-facing camera needs to have motion, as this is critical for generating COLMAP results, which our method relies on for input.
>
> 2.	Multi-view with Side-view GT: Each primary front-facing camera frame should include significantly deviated side-view images to serve as ground truth for calculating quantitative metrics such as PSNR and LPIPS.
>
> 3.	Dynamic Full-scene Representation: Since we render the entire scene, the dynamic dataset must also include background information.
>
> None of the existing datasets meet all three criteria detailed above, as shown in the comparison table below  (as well as Table S2 in the updated supplementary section). Even the NVIDIA dataset we tested in the paper, lacks the Side View GT; so while we do provide a qualitative comparison, a concrete quantitative comparison is simply impossible due to the limitations of the NVIDIA dataset. In fact, this is precisely why we created SynDM, to fill this gap and lay a solid foundation for future research.
>
> +-------------------+-----------------+---------------------------+---------------------------------+---------------------+--------------------+
>
> |----Dataset----|-Multi-view-|-Deviated-View GT-|-Unconstrained Scene-|-Cams Motion-|-Background-|
>
> +-------------------+-----------------+---------------------------+---------------------------------+---------------------+--------------------+
>
> |SynDM(Ours)|-------✔-------|-------------✔------------|----------------✔---------------|---------✔---------|---------✔---------|
>
> |-----DAVIS------|-------✘-------|-------------✘-----------|----------------✔---------------|---------✔---------|---------✔---------|
>
> |-----iPhone----|-------✘-------|-------------✘------------|----------------✔---------------|---------✔---------|---------✔---------|
>
> |----NeRFDS----|-------✘-------|-------------✘-----------|----------------✔----------------|---------✔---------|---------✔--------|
>
> |-----NVIDIA----|-------✘-------|-------------✘------------|----------------✔---------------|---------✔---------|---------✔--------|
>
> |--HyperNeRF-|-------✘-------|-------------✘------------|----------------✘---------------|---------✔---------|---------✔--------|
>
> |----DyNeRF----|-------✔-------|-------------✔------------|----------------✘---------------|---------✘---------|---------✔--------|
>
> |---ActorsHQ--|-------✔-------|-------------✔------------|----------------✘---------------|---------✘---------|---------✘--------|
>
> |---Multi face--|-------✔-------|-------------✔------------|----------------✘---------------|---------✘---------|---------✘--------|
>
> +-------------------+-----------------+---------------------------+---------------------------------+---------------------+--------------------+
>
> **Weakness Concern 2.2: "and do not demonstrate a clear advantage of the proposed method compared to other dynamic reconstruction methods"**
>
> -------------------------
>
> As shown in Figures 1, 5 and the supplementary video, our model achieves significantly better visual quality. Quantitatively, it consistently outperforms other methods in FID and LPIPS scores. These results clearly demonstrate the superior performance of our approach in dynamic scene reconstruction.
>
>
> **Weakness Concern 2.3: "Moreover, none of the chosen baselines utilizes a 3D prior, which seems unfair as the current solution leverages one."**
>
> --------------------------
>
> The 3D prior supervision is in fact a part of the novelty of our proposed model, and the characteristic that distinguishes our method from existing works. To clarify, the 3D prior is not an external input, but a feature derived from the shared input images. We provide no additional information exclusively to our model that would create any unfair standing in our evaluation, and all methods receive the same input images. All comparisons were conducted end-to-end with identical inputs, focusing solely on the final rendering results.

---

> ### Author Response · Authors · 2024-11-19
>
> **Weakness Concern 3: "Operational range [-45∘, 45∘] is rather limiting, especially for 360∘ object-centric dynamic reconstruction for creation of avatars and 4D assets. Addressing the issue appears important for the practical usage of this work"**
>
> ----------------------
>
> Our introduction of the SynDM dataset and ExpanDyNeRF is crucial in creating the groundwork required for practical applications, which were previously unattainable due to the lack of any comprehensive dataset with ground truth views. This work was in fact inspired by limitations faced within operational ranges, let alone full 360 reconstruction. By introducing a solution and dataset for this challenge, we lay the necessary step forward to achieving full reconstruction. While achieving fully dynamic 360-degree scene rendering remains an exciting goal and next step, even current sophisticated approaches like DreamGaussian4D currently operate within isolated, constrained settings with limited real-world applicability. Our work bridges this gap, making significant progress towards a robust, practical 360-degree dynamic reconstruction in future work.
>
> **Q1: "The concept of parallax described in Section 3.2 and motion perception are also addressed in [2] by leveraging semantic understanding. It would be interesting to include it as a baseline and test the method on the proposed significant rotational changes."**
>
> -------------------
>
> Including this method as a baseline would result in an unfair comparison, as our work tackles the more challenging and realistic scenario of dynamic full-scene rendering with limited monocular input. The referenced method [4] relies on a multi-view input and uses the ground truth from all deviated angles during training, which simplifies the task significantly. In contrast, our approach uses only a monocular input during training, with deviated-view ground truth reserved strictly for evaluation.
>
> [4] Liu et al. "Gear-NeRF: Free-Viewpoint Rendering and Tracking with Motion-aware Spatio-Temporal Sampling.", 2024.
>
> **Q2: "Can translational changes be addressed in the work, especially in the generation of novel deviated views? What about changes in the distance to the object (e.g. zoom-ins)?"**
>
> --------------------
>
> Our method can handle translational changes and variations (eg. zoom-in, dolly effect), but they are not the main focus and contribution of our work. These are basic evaluations that most dynamic models address successfully. Our primary contribution targets the more challenging problem of dynamic scene reconstruction under large, deviated angles, aiming to push the boundaries of current methodologies.
>
> **Q3: "Another potential benchmark could be ActorsHQ [3] or Multiface [4] for rendering dynamic humans and faces from various angles."**
>
> ---------------------
>
> Please refer to the response for Weakness 2.1.

---

### Official Review · Reviewer_p6LN · 2024-10-30

**Soundness:** 3
**Presentation:** 3
**Contribution:** 3
**Rating:** 6
**Confidence:** 4

**Summary:**

This paper aims to improve the modelling of dynamic scenes, particularly in terms of their quality when sampled from novel views. The authors introduce an Expanded Dynamic NeRF (ExpanDyNeRF) method for improved dynamic 3D scene reconstruction. Different from previous dynamic NeRFs, this method incorporates a 3D Gaussian Splatting prior into the pipeline, using it to supervise the training of a dynamic NeRF at each frame. The authors also curate a new synthetic dataset (SynDM) that has multiple viewpoints with dynamic camera motion, while also providing corresponding side views for evaluation purposes. On the proposed SynDM dataset. the proposed method achieves significantly better results as compared to existing dynamic scene reconstruction methods.

**Strengths:**

This paper tackles an important problem. Existing works that attempt to model dynamic scenes tend to perform badly at novel views. Thus, the tackled problem is meaningful.

This paper not only proposes a method for better modelling dynamic scenes, it also proposes a new synthetic dataset that may greatly facilitate the training of better models. In my opinion, the lack of a suitable dataset seems to be a fundamental issue blocking the progress of improvements in the area. This dataset may help to contribute to the long-term developments in the field.

The reported results on the proposed SynDM dataset, show that the proposed ExpanDyNeRF significantly surpasses the current SOTAs.

**Weaknesses:**

Although the proposed ExpanDyNeRF attains good performance, most of the improvements are largely engineering-based contributions without too many additional insights. For e.g., using a 3D gaussian prior and extracting novel views as supervision signals. Thus, since the technical novelty of the paper is not the best, and it seems there are no huge insights on why exactly the pipeline works well, the novelty of the paper may be limited.

The proposed method does not show gains on the existing NVIDIA dynamic scenes dataset. Even though the existing NVIDIA dataset only uses stationary cameras, and are not exactly the target of the proposed method, but the proposed method also uses a lot more components and computations, and thus should not offer lower performance than existing methods in my opinion.

Other specific concerns and questions have been placed under the “Questions” section.

**Questions:**

Firstly, it seems that the super-resolution loss (and the use of a pre-trained super-resolution model) is overly important. Specifically, from Table 2, it seems that most of the performance gains comes from the addition of the super-resolution loss L_sr. Furthermore, previous works mostly do not use a super-resolution loss or technique. Thus, this raises the possibility that the good performance and metrics of the method are mostly due to this pre-trained super-resolution model providing good guidance signals, instead of the other proposed designs (e.g., sampling from a 3D prior). Could the authors 1) provide more qualitative/quantitative results without the use of a pre-trained super-resolution method, or 2) provide experiment results on baseline methods, while adding the super-resolution loss to them. The results of either of these may show the actual extent of importance of the super-resolution loss, as well as better show the contributions of the other proposed designs.

Another concern lies in the optimization time. It seems like there are two large and potentially expensive components that have been added to the pipeline as compared to previous dynamic NeRF works: 1) Fitting of a 3D Gaussian Splatting prior at every frame; 2) Usage of a potentially large pre-trained super-resolution model for losses at every iteration. How much additional costs do these components add to the overall optimization time?


There are some claims that PSNR is not that good a metric for evaluating predictions with large deviation, as PSNR may not be sensitive to certain distortions. Although some qualitative evidence is provided in Fig 5 to show this, the evidence still seems a little lacking to dismiss the metric. For instance, it could be that D4NeRF may only have certain blurred frames (that have been shown), but otherwise performs quite well. Could there be further explanations or insights into why PSNR might fail here? Otherwise, could there be more qualitative visualizations to convince the reader? Since PSNR is often used to measure blurs and image quality, there needs to be more evidence or insights to make this claim convincing.


The authors should provide further clarifications on the dataset. From Section 3.3, it seems that there are 9 scenes, with 22 cameras each. How many of these cameras are moving? Also, how are these cameras moving? Are they all moving in the same way? At the same time, for each scene, is the scene recorded multiple times to feature different dynamic entities, e.g., humans? Otherwise, is each of the 9 scenes recorded with only one type of entity? Overall, how many scene/entity combinations are there?

---

> ### Author Response · Authors · 2024-11-19
>
> Thank you for your valuable feedback and recognition of our work. We will address your concerns and questions accordingly.
>
> **Weakness Concern: "The proposed method does not show gains on the existing NVIDIA dynamic scenes dataset. Even though the existing NVIDIA dataset only uses stationary cameras, and are not exactly the target of the proposed method, but the proposed method also uses a lot more components and computations, and thus should not offer lower performance than existing methods in my opinion."**
>
> The point you highlight stems from a tradeoff we make in our method, that ultimately improves the overall realism and reliability of our rendered scenes compared to other existing methods. Here is a detailed explanation of why this occurs.
>
> The NVIDIA dataset exhibits a certain bias in novel view evaluation. Its training set consists of 24 frames captured by 12 cameras. Specifically, it made up a monocular training set where each frame comes from a different camera at different timestamps. For instance, frame 1 is taken from camera 1, frame 2 from camera 2, and so on, until frame 13 loops back to camera 1. The novel view test still uses the same 24 camera positions, but at different timestamps.
>
> This setup allows the baseline NeRF, which optimizes only for front views, to have a highly comprehensive understanding of the scene information for both the training and novel views, resulting in better performance in front-view evaluations. In contrast, our method incorporates side-view constraints to improve side-view rendering quality. While these constraints make the side views significantly more realistic and complete, they introduce additional “noise” for pure front-view optimization. This trade-off leads to our method performing slightly worse in PSNR (3% lower in avg) than the baseline NeRF for front views.
>
> It is important to reiterate that this minor sacrifice in front-view performance is worthwhile, as our method demonstrates substantial improvements in side-view quality. This improvement greatly enhances the overall realism of the rendered scenes, highlighting the valuable advancement and contribution of our approach.

---

> ### Author Response · Authors · 2024-11-19
>
> **Q1: "Firstly, it seems that the super-resolution loss (and the use of a pre-trained super-resolution model) is overly important. Specifically, from Table 2, it seems that most of the performance gains comes from the addition of the super-resolution loss L_sr. Furthermore, previous works mostly do not use a super-resolution loss or technique. Thus, this raises the possibility that the good performance and metrics of the method are mostly due to this pre-trained super-resolution model providing good guidance signals, instead of the other proposed designs (e.g., sampling from a 3D prior). Could the authors 1) provide more qualitative/quantitative results without the use of a pre-trained super-resolution method, or 2) provide experiment results on baseline methods, while adding the super-resolution loss to them. The results of either of these may show the actual extent of importance of the super-resolution loss, as well as better show the contributions of the other proposed designs."**
>
> We have added more qualitative results to address your concern and compare the contributions of each loss function in our updated supplementary materials. Specifically, we direct you to Figure S3, where you can see the impact of incorporating different loss functions on rendering quality in our novel view synthesis. From the baseline, it is evident that severe tearing occurs in both the subject and the surrounding background. When using only the pre-trained super-resolution loss, it fails to address the tearing in the side-view perspectives, and the results remain unsatisfactory. However, when we incorporate both our novel view density loss and color loss, the images clearly show significant improvements: the subject and background are accurately reconstructed, with proper shapes and colors restored.
> It is important to note that the super-resolution loss was not introduced to resolve the side-view issues but rather to make the rendered results appear sharper and more visually appealing. This differentiation highlights the unique contributions of each loss function in our framework.
>
> **Q2: "Another concern lies in the optimization time. It seems like there are two large and potentially expensive components that have been added to the pipeline as compared to previous dynamic NeRF works: 1) Fitting of a 3D Gaussian Splatting prior at every frame; 2) Usage of a potentially large pre-trained super-resolution model for losses at every iteration. How much additional costs do these components add to the overall optimization time?"**
>
> For the fitting of the 3D Gaussian Splatting prior, it takes approximately 25–30 seconds to generate the pseudo ground truth for each frame, with a total of 24 frames in the dataset. The time cost of generating the Gaussian Splatting prior per frame is negligible compared to the time required for training NeRF.
>
> Here is a detailed breakdown of the overall optimization time with the pre-trained super-resolution model:
>
> •	The baseline D4NeRF requires an average of approximately 60 hours to train a single scene with 2 V100 GPU
>
> •	Incorporating our novel view density and color losses adds about 4 additional hours to the training process.
>
> •	Further incorporating the super-resolution loss increases the training time by an additional 8 hours.
>
> Training our ExpanDyNeRF model for a single scene takes approximately 72 hours. While these additions may slightly increase the overall training time, they are essential for improving model performance and achieving the high-quality results presented in our work.

---

> ### Author Response · Authors · 2024-11-19
>
> **Q3: "There are some claims that PSNR is not that good a metric for evaluating predictions with large deviation, as PSNR may not be sensitive to certain distortions. Although some qualitative evidence is provided in Fig 5 to show this, the evidence still seems a little lacking to dismiss the metric. For instance, it could be that D4NeRF may only have certain blurred frames (that have been shown), but otherwise performs quite well. Could there be further explanations or insights into why PSNR might fail here? Otherwise, could there be more qualitative visualizations to convince the reader? Since PSNR is often used to measure blurs and image quality, there needs to be more evidence or insights to make this claim convincing."**
>
> As shown in the heatmap results in Figure S4 (refer to the updated supplementary section of the paper), our model delivers much sharper and clearer outputs compared to the baseline, yet PSNR scores are slightly lower. This discrepancy arises because PSNR, derived from MSE, is sensitive to high-density errors in small, misaligned regions. Our sharper output may exhibit small localized misalignments, increasing MSE, while the blurry baseline has smoother transitions, resulting in lower MSE with inferior visual quality.
>
> This issue is particularly evident in areas where blurry regions blend with the background, such as the green area where human pants blur into the ground, minimizing MSE contributions. Thus, PSNR does not effectively capture the superior clarity and sharpness of our results, making it less suitable for fully evaluating our model's advantages.
>
> **Q4 "The authors should provide further clarifications on the dataset. From Section 3.3, it seems that there are 9 scenes, with 22 cameras each. How many of these cameras are moving? Also, how are these cameras moving? Are they all moving in the same way? At the same time, for each scene, is the scene recorded multiple times to feature different dynamic entities, e.g., humans? Otherwise, is each of the 9 scenes recorded with only one type of entity? Overall, how many scene/entity combinations are there?"**
>
> 1. How many of these cameras are moving? Are they all moving in the same way?
>
> All 22 cameras are moving, you can think of them as a giant flying dome, and they all move simultaneously. Each frame corresponds to images captured by all 22 cameras. The primary camera (used for training) is always the front-facing view, and the positions of the other cameras are determined relative to the front camera, as illustrated in Figure 3. All cameras move in a coordinated manner based on this configuration.
>
> 2. For each scene, is the scene recorded multiple times to feature different dynamic entities (e.g., humans)? Otherwise, is each of the 9 scenes recorded with only one type of entity? Overall, how many scene/entity combinations are there?
>
> For each object (e.g., human, animal, or vehicle), we use a dedicated set of dynamic movements and background scenes. Each object/scene combination is captured in a single recording. We do multiple captures to cover various objects/scenes but do not repeat recordings of the same background scene with different objects. This design avoids potential performance biases in comparisons due to repeated scenes.
>
> In total, there are 9 scenes, each featuring a distinct dynamic entity. Specifically:
>
> •	3 human scenes: male, female, zombie
>
> •	3 animal scenes: chicken, dog, and shark
>
> •	3 vehicle scenes: bus, firetruck and monster truck.
>
> We do not capture variations within the same background scene, such as using different human characters in the same environment, as we believe this could introduce unintended biases in the dataset.

---

> > ### Comment · Reviewer_p6LN · 2024-11-24
> >
> > I would like to thank the authors for their responses, which have clarified some details and some of my concerns.
> >
> > However, I remain unconvinced regarding several aspects. For instance, the performance on the NVIDIA dynamic scenes dataset is not that good, even though the proposed method adds additional significant modules upon previous works. Furthermore, the authors only experiment on this public dataset, and there are other mainstream datasets that have not been experimented on, which means these results may not hold up well on other datasets.
> >
> > Additionally, I am not very convinced that the 3D prior is the factor that provides significant gains, even after looking at the qualitative results. This is because the quantitative gains by the super-resolution module are very significant compared to the 3D prior, and explaining the strength of the 3D prior despite this discrepancy is difficult just by the visualization of a single sample.
> >
> > As these concerns remain, I choose to keep my score.

---

> > > ### Author Response · Authors · 2024-11-28
> > >
> > > Thank you for your recognition of our previous responses, and we would like to address your comments and suggestions in detail.
> > >
> > > **However, I remain unconvinced regarding several aspects. For instance, the performance on the NVIDIA dynamic scenes dataset is not that good, even though the proposed method adds additional significant modules upon previous works.**
> > >
> > > As shown in Figures S1, S8, and S9, other methods exhibit issues such as scattering, disjointed reconstructions, blurriness, or a cardboard-like effect on objects. In contrast, our method delivers sharp and clear renderings while maintaining depth consistency, even under deviated views.
> > >
> > > **Furthermore, the authors only experiment on this public dataset, and there are other mainstream datasets that have not been experimented on, which means these results may not hold up well on other datasets.**
> > >
> > > We recognize the importance of evaluating our method on a broader range of datasets. However, existing monocular video datasets, such as iPhone and NeRF-DS, lack multiview frames with significant viewpoint changes at each timestamp due to their inherently monocular nature. While we initially considered using camera poses from temporally distant frames as novel views, this approach proved ineffective. In dynamic environments, objects are in motion, meaning frames captured at later times—even from different viewpoints—fail to represent the same scene at the same moment as the original frame. Using such frames as ground truth introduces temporal inconsistencies, compromising reconstruction accuracy.
> > >
> > > Our goal is to reconstruct dynamic scenes where the background plays a vital role. Multiview datasets like ActorsHQ and Multiface lack backgrounds, making them unsuitable for our study. Meanwhile, other multiview datasets with backgrounds, such as NVIDIA and DyNeRF, are captured using fixed camera matrices. To synthesize monocular input from these datasets, we would need to extract frames from different cameras at varying time points to simulate camera motion, enabling COLMAP to compute camera poses. However, this synthetic camera motion introduces discontinuities and optical flow noise between frames during training. To overcome these limitations, we proposed the SynDM dataset, specifically designed to provide reasonable perspectives for evaluating models and addressing gaps in the field.
> > >
> > > While we considered simulating a monocular video with camera motion using multiview datasets like DyNeRF by selecting frames from different cameras across consecutive time steps, this approach was debatable. The deviated views introduced during this imitation process would provide additional information during training, leading to unfair comparisons. Our analysis of cameras in DyNeRF, such as cam0, cam4, and cam5—relatively close to the front view—revealed that their deviation range is approximately [-10°, +10°]. However, our study shows that deviations exceeding 10° result in obvious blur and artifacts (Figure S7, S8), which means the simulated monocular training video from DyNeRF provide deviated view ground truth during training.
> > >
> > > To avoid potential criticisms for using side-view data to train a monocular-based method and to better showcase the advantages of our approach that can reconstruct the scene without the information from side views, we focused solely on the NVIDIA dataset, where all camera are aligned and have no deviation between cameras, and the SynDM dataset, where the camera pose deviation between adjacent frames is under 1°.
> > >
> > > **Additionally, I am not very convinced that the 3D prior is the factor that provides significant gains, even after looking at the qualitative results. This is because the quantitative gains by the super-resolution module are very significant compared to the 3D prior, and explaining the strength of the 3D prior despite this discrepancy is difficult just by the visualization of a single sample.**
> > >
> > > Regarding your doubts about whether the improvements in rendering results are primarily due to the 3D prior or the super-resolution module, we have provided further clarification in the supplementary material, specifically in Figure S3. From this figure, it is evident that a model optimized using only the super-resolution loss produces "high-resolution" results that are still inherently blurry. This occurs because the baseline output is too blurry to enable correct feature matching during training.
> > >
> > > In our method, the novel view loss plays a crucial role in optimizing both density and color, while the super-resolution module focuses solely on further refining the color. As such, novel view loss is essential for improving the overall quality of the reconstruction. Without the 3D prior's contribution via novel view loss, the optimization process would lack the necessary supervision for density and structural information, resulting in unsatisfactory outcomes.

---

### Official Review · Reviewer_aYsa · 2024-10-31

**Soundness:** 3
**Presentation:** 2
**Contribution:** 2
**Rating:** 5
**Confidence:** 5

**Summary:**

The paper introduces ExpanDyNeRF, a method that enhances dynamic Neural Radiance Field (NeRF) models for rendering 3D scenes from viewpoints significantly deviated from the primary camera. Traditional dynamic NeRFs struggle with these novel views, often producing artifacts or blurring. ExpanDyNeRF incorporates a Gaussian splatting prior and a pseudo ground truth approach, which help refine density and color features, enabling realistic scene reconstructions from various perspectives. The paper also presents SynDM, a new GTA V-based multiview dataset specifically designed for dynamic scene reconstruction with diverse angles. Through quantitative and qualitative evaluations on SynDM and the NVIDIA dataset, ExpanDyNeRF shows superior performance in rendering stability and fidelity compared to other state-of-the-art models, addressing limitations in existing NeRF methods.

**Strengths:**

1. A SynDM dataset, a novel GTA V-based dynamic Multiview dataset. This dataset provides a valuable resource for evaluating dynamic 3D reconstructions from varying viewpoints, with ground truth that supports assessment of model generalization to wide-angle novel views.

2. To enhance dynamic Neural Radiance Field (NeRF) models for rendering 3D scenes from viewpoints significantly deviated from the primary camera.

**Weaknesses:**

1. The process for generating pseudo-ground truth is somewhat unclear. The way to use Gaussian splatting prior is not clear. Further explanation on how this method connects would clarify the methodology.

2. The integration of Gaussian splatting and its specific role in improving novel view synthesis lacks sufficient clarity. Some recent research such as [1] deploy diffusion as prior has achieved great results. Additional details on how Gaussian splatting affects density and color optimization at deviated angles would strengthen the understanding of its contribution to model performance.

3. The experiments only demonstrate performance on the SynDM and NVIDIA datasets. I suggest conducting experiments on datasets more widely used in dynamic NeRF, such as the DAVIS and iPhone datasets.

[1] Chen, H., Loy, CC, & Pan, X. (2024). MVIP-NeRF: Multi-view 3D Inpainting on NeRF Scenes via Diffusion Prior. In Proceedings of the IEEE/CVF Conference on Computer Vision and Pattern Recognition (CVPR), 5344-5353.

**Questions:**

In the discussion about the GTA-V platform, the author refers “semi-freezing" of the game’s graphic state. How does it differ from a complete freeze and how does it enable smooth transitions across multiple camera views without introducing significant latency?

---

> ### Author Response · Authors · 2024-11-19
>
> Thank you for your valuable feedback and recognition of our work. We will address your concerns and questions accordingly.
>
> **Q1. "In the discussion about the GTA-V platform, the author refers “semi-freezing" of the game’s graphic state. How does it differ from a complete freeze and how does it enable smooth transitions across multiple camera views without introducing significant latency?"**
>
> In the native game state, running GTA V smoothly and producing ideal rendering frames requires both a physics engine (responsible for executing actor scripts and swap function calls) and a rendering engine (responsible for updating image frames based on those function calls).
>
> In the native pause (“complete freeze”) function, both the physics and rendering engines cease operation. Conversely, in the pseudo pause (“semi-freeze”) function, only the rendering engine stops working, while the physics engine continues operating in the background.
>
> To provide a rough idea of how we achieve smooth transitions with minimal latency, let’s compare these two swapping methods step by step:
>
> Native Pause-Based Swap (complete freeze)
>
> 1.	Step 0: Display the current frame from camera 1.
>
> 2.	Step 1: Call the native pause function. This freezes both the physics and rendering engines, storing camera 1’s image frame and metadata.
>
> 3.	Step 2: Release the native pause, unfreezing both the physics and rendering engines.
>
> 4.	Step 3: The physics engine processes the camera swap function call.
>
> 5.	Step 4: The rendering engine updates to display camera 2’s image frame.
>
> 6.	Step 5: Native pause the system again to store camera 2’s image frame and metadata, then repeat the process.
>
> Pseudo Pause-Based Swap (semi-freeze)
>
> 1.	Step 0: Display the current frame from camera 1.
>
> 2.	Step 1: Call the pseudo pause function. This freezes only the rendering engine, allowing the physics engine to remain active. At this step, we simultaneously store camera 1’s image frame and metadata while the physics engine continue processes actor scripts and the camera swap function call.
>
> 3.	Step 2: Release the pseudo pause, unfreezing the rendering engine. The rendering engine immediately reacts to the swap function call and displays camera 2’s image frame.
>
> 4.	Step 3: Pseudo Pause the system again to store camera 2’s image frame and metadata, then repeat the process.
>
> Comparison
>
> As shown in this step-by-step explanation, the pseudo pause-based swap successfully eliminates the most time-consuming steps. On a 60Hz display, the latency is reduced from 0.0167s–0.033s (native pause) to approximately 0.002s–0.003s (pseudo pause), representing an 80- to 100-fold reduction.
>
> This significant decrease in latency makes the pseudo pause-based swap particularly effective for depth-related tasks, ensuring smooth transitions with minimal delays.

---

> ### Author Response · Authors · 2024-11-19
>
> **Weakness concerns 1&2:**
>
> **"The process for generating pseudo-ground truth is somewhat unclear. The way to use Gaussian splatting prior is not clear. Further explanation on how this method connects would clarify the methodology."**
>
> **"The integration of Gaussian splatting and its specific role in improving novel view synthesis lacks sufficient clarity. Some recent research such as [1] deploy diffusion as prior has achieved great results. Additional details on how Gaussian splatting affects density and color optimization at deviated angles would strengthen the understanding of its contribution to model performance."**
>
> ------------------------------------------------------------------------------------------------------------------------------
>
> Our pseudo-ground truth generation involves leveraging Gaussian splatting to construct a 3D reconstruction mesh for the foreground object in each frame, which is the gaussian prior we referred in the paper. Specifically, we generate the shape and texture pseudo-ground truth for the foreground object by capturing novel views, as outlined in Figure 3. This process enables us to optimize density and color for novel viewpoints with greater consistency and precision across frames.
>
> Gaussian splatting is particularly suitable for this context due to its balance between training efficiency and rendering quality. This allows us to generate pseudo-ground truth efficiently while preserving high geometric and textural fidelity. During our experiments, we evaluated various Gaussian splatting approaches, including DreamGaussian, Stable-Zero123, and One-2-3-45. Ultimately, DreamGaussian was selected for its superior performance and computational efficiency.
>
> Regarding your mention of MVIP-NeRF, it is a multi-view-based method, inherently benefiting from direct feed of side view GT information compared to our monocular-focused ExpanDyNeRF. Tackling the challenges of monocular input presents additional complexity,
>
> Furthermore, it is also important to note that it is not a method specifically focused on dynamic scenes. However, I understand your point—they utilize diffusion priors to achieve supervision, and we appreciate the comparison.
>
> In fact, we initially explored a similar approach, aiming to use diffusion priors for pseudo-ground truth generation. However, we observed that diffusion priors lack the necessary precision to serve as reliable pseudo-ground truth. The inherent randomness in image rendering for target camera poses introduces small but critical inconsistencies, which can impede training convergence and final rendering quality.
>
> In contrast, our Gaussian splatting approach provides a fixed 3D mesh prior, ensuring significantly better consistency when rendering target camera poses. This consistency facilitates successful model convergence.

---

> ### Author Response · Authors · 2024-11-19
>
> **Weakness Concern 3: "The experiments only demonstrate performance on the SynDM and NVIDIA datasets. I suggest conducting experiments on datasets more widely used in dynamic NeRF, such as the DAVIS and iPhone datasets."**
>
> -----------------------------
>
> It is important to note that it is not a matter of us choosing whether not to use these datasets, but rather that the fact that they do not meet the necessary conditions for a fair comparison with our method.
>
> Specifically, for a dataset to be used for high quality evaluation of the ExpanDyNeRF, it must the folllowing three key requirements:
>
> 1.	Monocular input with motion: The primary front-facing camera needs to have motion, as this is critical for generating COLMAP results, which our method relies on for input.
>
> 2.	Multi-view with Side-view GT: Each primary front-facing camera frame should include significantly deviated side-view images to serve as ground truth for calculating quantitative metrics such as PSNR and LPIPS.
>
> 3.	Dynamic Full-scene Representation: Since we render the entire scene, the dynamic dataset must also include background information.
>
> There are no well-established datasets that meet all three requirements. To illustrate this, we compared existing datasets (monocular and multi-view stereo) against our SynDM dataset, as shown in the comparison table below (as well as Table S2 in the updated supplementary section). This gap highlights the shortcomings of SOTA datasets in fulfilling these critical conditions to enable further research.
>
> +-------------------+-----------------+---------------------------+---------------------------------+---------------------+--------------------+
>
> |----Dataset----|-Multi-view-|-Deviated-View GT-|-Unconstrained Scene-|-Cams Motion-|-Background-|
>
> +-------------------+-----------------+---------------------------+---------------------------------+---------------------+--------------------+
>
> |SynDM(Ours)|-------✔-------|-------------✔------------|----------------✔---------------|---------✔---------|---------✔---------|
>
> |-----DAVIS------|-------✘-------|-------------✘-----------|----------------✔---------------|---------✔---------|---------✔---------|
>
> |-----iPhone----|-------✘-------|-------------✘------------|----------------✔---------------|---------✔---------|---------✔---------|
>
> |----NeRFDS----|-------✘-------|-------------✘-----------|----------------✔----------------|---------✔---------|---------✔--------|
>
> |-----NVIDIA----|-------✘-------|-------------✘------------|----------------✔---------------|---------✔---------|---------✔--------|
>
> |--HyperNeRF-|-------✘-------|-------------✘------------|----------------✘---------------|---------✔---------|---------✔--------|
>
> |----DyNeRF----|-------✔-------|-------------✔------------|----------------✘---------------|---------✘---------|---------✔--------|
>
> |---ActorsHQ--|-------✔-------|-------------✔------------|----------------✘---------------|---------✘---------|---------✘--------|
>
> |---Multi face--|-------✔-------|-------------✔------------|----------------✘---------------|---------✘---------|---------✘--------|
>
> +-------------------+-----------------+---------------------------+---------------------------------+---------------------+--------------------+
>
>
>
>
> For example, while the real-world NVIDIA dataset was included in our qualitative comparisons, the lack of side-view ground truth (GT) prevents quantitative evaluation. Similarly, datasets like DAVIS and the iPhone fail to provide the necessary side-view GT, making them unsuitable for producing quantitative results. This underscores the unique value of our SynDM dataset in addressing this gap and advancing future research in the field.

---

> > ### Comment · Reviewer_aYsa · 2024-11-24
> > **Reply to authors' rebuttal**
> >
> > Thanks for the authors' rebuttal and it addresses some of my concerns.
> >
> > However, the main concern is still unaddressed: the lack of experiments on widely used datasets. Although the authors point out that these datasets lack side view GT, I think this problem can be alleviated by increasing the frame interval even if the camera transformation between adjacent frames is small. In addition, I think the goal of this work should focus on the reconstruction of the main view, and the role of the side view should be to improve the reconstruction quality of the main view. Because in real applications, most wild videos are monocular, and there is a lack of video resources that specifically provide side views, the application scenarios are relatively limited. Therefore, it is recommended that the authors consider verification on more widely used datasets, or further demonstrate the potential value of their methods in practical applications. So I keep my score.

---

> > > ### Author Response · Authors · 2024-11-28
> > >
> > > Thank you for recognizing our previous responses. We appreciate your constructive feedback, and we would like to address your comments and suggestions in detail.
> > >
> > > **"I think this problem can be alleviated by increasing the frame interval even if the camera transformation between adjacent frames is small,"**
> > >
> > >  we appreciate the idea but find it challenging to implement effectively. Suppose your suggestion implies using a larger frame interval during evaluation to obtain the side view ground truth, while increasing the frame interval might seem like a solution to address small camera transformations between adjacent frames. In that case, this approach is impractical for reconstructing dynamic scenes from monocular video. In dynamic environments, objects are moving, so frames captured at later times—even from different viewpoints—do not represent the same scene at the same moment as the original frame. Therefore, using frames from a larger interval as ground truth for side views would introduce temporal inconsistencies due to the motion of objects, undermining the reconstruction accuracy. Moreover, this method would require specific camera movements, such as circling around the scene, which imposes additional constraints on data collection and limits the generalizability of our approach. Our goal is to develop a model that can be trained using general monocular inputs without such restrictions, ensuring broader applicability across various types of dynamic scenes.
> > >
> > > If you mean increasing the frame interval during the training, it could lead to significant discrepancies in object states between adjacent frames, particularly for fast-moving objects. This would amplify flow loss and hinder the establishment of temporal consistency between frames, potentially resulting in artifacts such as missing parts of objects during training. Maintaining consistency across frames is critical for producing high-quality results, and increasing the frame interval would compromise this. I hope my response has addressed your concerns. If I misunderstood anything, please feel free to ask further questions.
> > >
> > >
> > > **“In addition, I think the goal of this work should focus on the reconstruction of the main view, and the role of the side view should be to improve the reconstruction quality of the main view. Because in real applications, most wild videos are monocular, and there is a lack of video resources that specifically provide side views, the application scenarios are relatively limited. ”**
> > >
> > > As for your comments on the core motivation of our work, we emphasize the challenge identified in the introduction: *"However, novel view renderings often appear blurry and filled with artifacts, especially when significantly deviating from the primary camera view. This observation, shown in Fig. 1, is understandable since both methods lack supervision from diverse views while training, a limitation inherent to monocular camera settings. To address this challenge, we introduce Expanded Dynamic NeRF (ExpanDyNeRF)."*
> > >
> > > Our primary goal is to address the limitations of reconstructing 3D dynamic scenes from monocular videos with constrained viewpoints. If methods focus only on optimizing views appearing in the training videos, the results often resemble the 4DGS case we presented, where objects perform well in the primary view but suffer from disjointing and scattering inside views due to overfitting. ExpanDyNeRF aims to overcome these limitations by enabling robust reconstruction for a wider range of views.
> > >
> > > **"Because in real applications, most wild videos are monocular, and there is a lack of video resources that specifically provide side views, the application scenarios are relatively limited."**
> > >
> > > We agree that most wild videos are indeed monocular, making multi-view training methods less feasible. Furthermore, in scenarios such as fixed audience seating at sports events and concerts or when collecting wildlife data, obtaining monocular videos with significant camera pose changes is often impractical. Our method is designed to address precisely these scenarios, enabling the reconstruction of 3D scenes from monocular videos that provide clear and sharp renderings across a broader range of views.
> > >
> > > One impactful application derived from this research is its use in educational settings, such as virtual field trips for students with disabilities or socio-economic challenges who may not be able to attend in-person visits to science centers or other locations. By expanding their view and enhancing their experience beyond a single monocular perspective, ExpanDyNeRF has the potential to make learning more immersive and inclusive.
> > >
> > > Thank you again for your thoughtful feedback. We hope our clarifications adequately address your concerns and demonstrate the significance and potential of our work.

---

### Official Review · Reviewer_uC2Y · 2024-11-02

**Soundness:** 3
**Presentation:** 3
**Contribution:** 3
**Rating:** 6
**Confidence:** 3

**Summary:**

This paper introduces ExpanDyNeRF, an innovative approach to dynamic NeRFs for novel view synthesis that tackles large-angle viewpoint deviations. By leveraging Gaussian splatting and pseudo-ground truth optimization, the method achieves notable stability and realism in dynamic scenes, even from challenging perspectives. A key contribution is the introduction of the Synthetic Dynamic Multiview (SynDM) dataset, built from GTA V, specifically designed to evaluate dynamic, multi-view scenes—filling a critical gap in the current dataset landscape. ExpanDyNeRF demonstrates significant improvements over state-of-the-art methods, with substantial gains in FID, LPIPS, and PSNR metrics, particularly excelling in color and shape fidelity.

**Strengths:**

Comparisons with state-of-the-art dynamic multi-view synthesis methods show clear improvements both quantitatively and qualitatively, especially in larger viewpoint settings. The proposed pseudo-ground truth optimization with Gaussian priors enhances scene details, with each component thoroughly evaluated in ablation studies. The SynDM dataset, offering multi-view dynamic data with ground-truth side views for benchmarking, is a valuable addition to the community, enabling reproducible evaluations in dynamic NeRF research. Efforts to reduce latency are also commendable, and making the source code public would further benefit the field.

**Weaknesses:**

While benchmarking on synthetic data is understandable given dataset constraints, additional evaluations on real-world data would more thoroughly demonstrate the method’s robustness. Handling larger viewpoint deviations remains challenging, as both the baseline and proposed methods show limitations. A discussion of promising future directions—particularly whether this approach opens up possibilities for substantial gains in this area—would be valuable. Furthermore, an analysis of runtime and computational resource usage would help assess the scalability and deployment potential of the method.

**Questions:**

1. Have you considered testing ExpanDyNeRF on real-world dynamic datasets? How do you anticipate the model would perform on non-synthetic data, and are there plans to include such evaluations in future work?

2. Both the baseline and proposed methods encounter limitations with larger viewpoint deviations. Do you foresee any specific improvements or techniques that could help address these challenges? Does ExpanDyNeRF open up any new avenues for significant performance gains in this area?

3. Could you provide more details on the runtime performance and computational resources required for both training and inference? Understanding the method’s efficiency and memory usage would help assess its scalability and real-world deployment feasibility.

4. Given the engineering efforts noted in reducing latency, do you plan to release the source code for ExpanDyNeRF? This would be valuable for the community and help facilitate reproducible research.

---

> ### Author Response · Authors · 2024-11-19
>
> Thank you for your valuable feedback and recognition of our work. We will address your concerns and questions accordingly.
>
> **Q1. Have you considered testing ExpanDyNeRF on real-world dynamic datasets? How do you anticipate the model would perform on non-synthetic data, and are there plans to include such evaluations in future work?**
>
> ------------------------------------------------
>
> Yes, we did test ExpanDyNeRF on the real-world NVIDIA dataset and have provided qualitative evaluations in the paper. However, like the NVIDIA dataset, most well-known real-world datasets lack the necessary requirements for a fair quantitative evaluation for our method.
>
> Specifically, for a high quality evaluation of ExpanDyNeRF, a dataset must meet these three key requirements:
>
> 1.	Monocular input with motion: The primary front-facing camera needs to have motion, as this is critical for generating COLMAP results, which our method relies on for input.
>
> 2.	Multi-view with Side-view GT: Each primary front-facing camera frame should include significantly deviated side-view images to serve as ground truth for calculating quantitative metrics such as PSNR and LPIPS.
>
> 3.	Dynamic Full-scene Representation: Since we render the entire scene, the dynamic dataset must also include background information.
>
> There are no well-established datasets that meet all these three criteria, real-world or not. Given this scenario, we still compared the existing datasets against our SynDM dataset, and the results highlight the shortcomings of real-world existing datasets in fulfilling these conditions.
>
>
> +-------------------+-----------------+---------------------------+---------------------------------+---------------------+--------------------+
>
> |----Dataset----|-Multi-view-|-Deviated-View GT-|-Unconstrained Scene-|-Cams Motion-|-Background-|
>
> +-------------------+-----------------+---------------------------+---------------------------------+---------------------+--------------------+
>
> |SynDM(Ours)|-------✔-------|-------------✔------------|----------------✔---------------|---------✔---------|---------✔---------|
>
> |-----DAVIS------|-------✘-------|-------------✘-----------|----------------✔---------------|---------✔---------|---------✔---------|
>
> |-----iPhone----|-------✘-------|-------------✘------------|----------------✔---------------|---------✔---------|---------✔---------|
>
> |----NeRFDS----|-------✘-------|-------------✘-----------|----------------✔----------------|---------✔---------|---------✔--------|
>
> |-----NVIDIA----|-------✘-------|-------------✘------------|----------------✔---------------|---------✔---------|---------✔--------|
>
> |--HyperNeRF-|-------✘-------|-------------✘------------|----------------✘---------------|---------✔---------|---------✔--------|
>
> |----DyNeRF----|-------✔-------|-------------✔------------|----------------✘---------------|---------✘---------|---------✔--------|
>
> |---ActorsHQ--|-------✔-------|-------------✔------------|----------------✘---------------|---------✘---------|---------✘--------|
>
> |---Multi face--|-------✔-------|-------------✔------------|----------------✘---------------|---------✘---------|---------✘--------|
>
> +-------------------+-----------------+---------------------------+---------------------------------+---------------------+--------------------+
>
>
>
> To illustrate our point, we provide the comparison table above (as well as the Table S2 in the updated supplementary section). As shown, none of the real-world datasets we examined simultaneously satisfy all three requirements necessary for a fair evaluation. This limitation explains why our evaluation on real-world datasets is constrained. Given this gap we have noticed through our research, we plan to create a real-world dataset ourselves. This endeavor will involve constructing a large-scale moving camera dome equipped with precise pose capture capabilities.

---

> ### Author Response · Authors · 2024-11-19
>
> **Q2. "Both the baseline and proposed methods encounter limitations with larger viewpoint deviations. Do you foresee any specific improvements or techniques that could help address these challenges? Does ExpanDyNeRF open up any new avenues for significant performance gains in this area? "**
>
> -----------------------------
>
> Yes, we do have plans to further enhance performance. Currently, the limitations stem from two main aspects: maintaining shape and color consistency for dynamic foreground objects and addressing background information loss at deviated angle. While our ExpanDyNeRF effectively tackles the first issue by ensuring temporal and structural consistency in dynamic objects, the next step is to leverage diffusion-based 3D scene generation techniques to reconstruct and fill in the missing background details. This comprehensive approach aims to ultimately address these challenges and further improve performance.
>
> ExpanDyNeRF indeed opens up new avenues for research and development in this field. For example, it provides a unified framework for dynamic full-scene rendering that ensures both temporal consistency and spatial scaling accuracy. This can inspire future work to focus on integrating diffusion models for seamless background generation or adaptive scene reconstruction techniques that perform well even with significant viewpoint deviations. Additionally, its flexibility could enable applications such as virtual reality (VR) environments, where users require immersive and coherent dynamic scenes with wide viewing angles.
>
>
> **Q3: "Could you provide more details on the runtime performance and computational resources required for both training and inference? Understanding the method’s efficiency and memory usage would help assess its scalability and real-world deployment feasibility. "**
>
> -------------------------
>
> ExpanDyNeRF requires approximately 72 hours to complete 600,000 iterations for a 24-frame input video for an entire training pipeline (including pseudo ground truth generation and target scene training), using 2 V100 GPU. In comparison, our baseline model, D4NeRF, takes 60 hours to train the same scene.
>
>
> **Q4: "Given the engineering efforts noted in reducing latency, do you plan to release the source code for ExpanDyNeRF? This would be valuable for the community and help facilitate reproducible research."**
>
> -------------------------
>
> Yes, we plan to release the complete source code for ExpanDyNeRF along with the resulting SynDM dataset. We are delighted to contribute to the research community and hope this will support reproducible research while advancing progress in the field.

---

> > ### Comment · Reviewer_uC2Y · 2024-11-24
> > **Rebuttal Comment**
> >
> > Thanks for the authors' rebuttal and it addresses some of my concerns.
> >
> > I appreciate the detailed explanation of the dataset and the future directions. However, the efficiency of the proposed method is quite a bottleneck and I couldn't see a promising solution based on the current process. I will keep my current score.

---

> > > ### Author Response · Authors · 2024-11-27
> > >
> > > Thank you for recognizing our previous responses!
> > >
> > > We acknowledge that training speed is a common challenge in dynamic NeRF research. For instance, DynIBaR: Neural Dynamic Image-Based Rendering (CVPR 2023) reported training times of approximately 48 hours, as stated in Section 4, Implementation Details: "Optimizing a full system on a 10-second video takes around two days using 8 Nvidia A100 GPUs." Similarly, MonoNeRF: Learning a Generalizable Dynamic Radiance Field from Monocular Videos (CVPR 2023) required approximately 50 hours of training on a single Nvidia A100 GPU.
> > >
> > > While training efficiency is undoubtedly important for real-time applications, ExpanDyNeRF is NOT designed for real-time usage at this stage. Instead, the primary contribution of our work lies in proposing a novel approach for expanded-view video reconstruction. This has significant benefits for many offline applications, such as enhancing the field of view of monocular recordings in entertainment or educational settings.
> > >
> > > In short, the speed of our method is highly dependent on the hardware used for training. Based on GPU benchmark data from Lambda Labs (\url{https://lambdalabs.com/blog/nvidia-a100-vs-v100-benchmarks}), a single A100 GPU is 2.17 times faster than a V100, while 8 A100 GPUs are 15.81 times faster than a single V100. Considering that our system uses V100 GPUs, our training speed is well within a reasonable and competitive range for dynamic NeRF research.

---

### Note · Authors · 2025-01-31

I have read and agree with the venue's withdrawal policy on behalf of myself and my co-authors.